# Distilling Causal Signals for One-Shot Directed Evolution of Antibodies

**Sai Pooja Mahajan**[1,†]    **Natasa Tagasovska**[1,†]    **Stefania Vasilaki**[1]    **Arian Rokkum Jamasb**[1]
**Andrew Martin Watkins**[1]    **Rajesh Ranganath**[2]

[†] equal contribution    [1]AIDD, Genentech    [2]Courant Institute of Mathematical Sciences, NYU
mahajs17@gene.com

## Abstract

Improving antibody binding to an antigen without antibody–antigen complex structures or antigen-specific training data is a central challenge in therapeutic protein design. We introduce **AffinityEnhancer**, a framework for one-shot antibody affinity improvement with strong generalization: given a single lead sequence, we propose variants that increase affinity without fine-tuning on the lead and without using antigen information, epitope/paratope labels, or the lead's structure in complex with the antigen. During training, AffinityEnhancer leverages a pan-antigen dataset of diverse binding environments (antigens) and constructs paired examples of related sequences with higher vs. lower measured binding. A shared, structure-aware module learns to transform low-affinity sequences toward high-affinity ones, distilling consistent, causal features associated with improved binding across environments. By combining pretrained sequence–structure embeddings with a sequence decoder, AffinityEnhancer generalizes to entirely unseen antibody seeds. Across multiple held-out internal and public leads, AffinityEnhancer concentrates mutations on the rim of the paratope, outperforms existing structure-conditioned and inpainting baselines, and achieves substantial *in silico* affinity gains in true one-shot experiments, despite never observing antigen-specific data at test time.[https://github.com/prescient-design/AffinityEnhancer]

## 1 Introduction

Antibodies are proteins produced by the immune system in response to foreign antigens. In therapeutic settings, antibodies have been developed as drugs against various cancer and autoimmune targets. They detect harmful antigens (such as bacteria and viruses) by the mechanism of *binding*, attaching to a specific patch on the antigen's surface, called an *epitope*, using six hypervariable loops known as complementarity-determining regions (CDRs). A subset of the residues on these CDRs form the antigen binding surface is known as the *paratope*.

This ability to form highly specific paratopes which are complementary in shape and chemical composition to a extensive repertoire of antigens confers unique therapeutic potential and makes high-affinity antibodies prime drug candidates. Having the therapeutic potential being driven by the binding mechanism, renders structure information as essential in developing solutions for this task. In the typical drug discovery pipeline, a lead antibody with reasonably high affinity and specificity to the antigen of interest, is identified from immunized libraries extracted from animals, followed by optimizing the lead for potency and drug-like properties. Optimizing the potency of the lead routinely involves improving its binding or affinity to the antigen. This process is called *affinity maturation*. Experimentally, affinity maturation involves random or directed mutagenesis to generate large diversified libraries (known as diversification or hit-expansion) followed by screening for stronger binding antibodies against the target. Such techniques are common in drug discovery pipelines and have been fairly successful over the last few decades. However, such diversification explores only a minuscule sequence space ($\sim$ order of $\sim 10^6$-$10^9$) of the entire sequence space (order of $250^{20}$; 20 amino acid residues at every position of the variable domain which consists roughly of

250 residues). As a consequence, the resulting sets of designs can be suboptimal and fail to identify sufficient number of antibodies with the desired potency and drug-like properties.

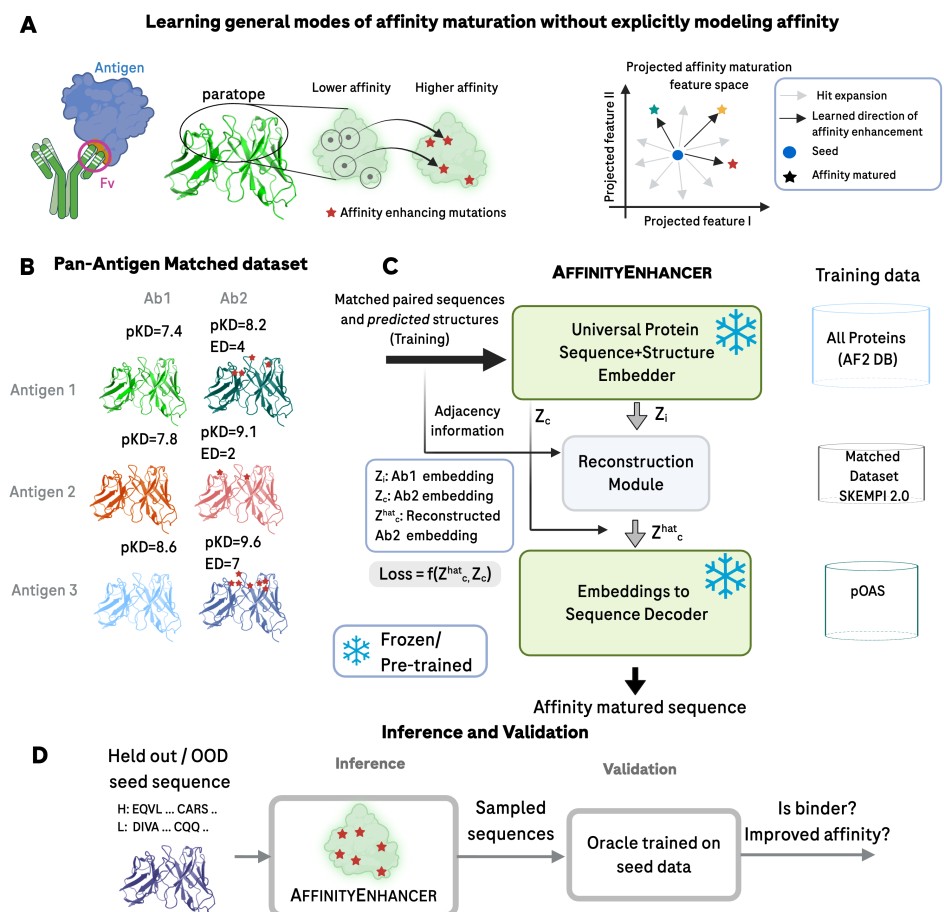

Figure 1: One-shot affinity maturation of antibodies with AFFINITYENHANCER. A) The goal is to implicitly learn modes of affinity maturation by pairing a lower affinity antibody with a higher affinity one. B) Matched datasets are obtained by pairing antibodies against the same target/antigen from the SKEMPI 2.0 database. C) Architecture for AFFINITYENHANCER. D) Inference and validation pipeline for held-out-seed to determine whether sampled sequences are binders or not.

Computational affinity maturation with machine learning offers an accelerated alternative to random or directed mutagenesis. However, the one-shot setting—where a model must propose improved variants from a single lead sequence without antigen context or fine-tuning—poses a key generalization challenge: the lead may be far from the training distribution in sequence and structural features. This challenge is compounded by the limited availability and diversity of paired antibody–antigen structures and affinity measurements, which impedes robust transfer to unseen targets (Hummer et al., 2023).

To bypass the challenges associated with explicitly modeling affinity, Tagasovska et al. (2024) proposed Property Enhancer (PropEn), a property-agnostic model which utilizes data matching to implicitly learn the direction of the gradient for a property of interest with the goal of proposing new optimized designs. It was previously demonstrated that this approach works for a range of tasks, including affinity maturation of antibodies. However, its effectiveness was only demonstrated in sequence-based models and in cases where a few hundred sequences related to the lead molecule we wish to optimize are already available in the training data, hence, not suitable to one-shot scenarios. In this work, we propose AFFINITYENHANCER, a model that goes beyond the PropEn framework, namely to the one-shot affinity maturation setup by leveraging structure information and introducing

a novel, diversified matching procedure which allows for generalization and transferability. Our main contributions are as follows:

- We propose a one-shot model for affinity maturation *without antigen information* (section 3)
- We leverage matching in heterogeneous datasets to bolster data-sparse regimes (antibody-antigen interactions)
- We provide theoretical analysis supporting distillation and OOD transfer of causal features (subsection 3.1)
- On held-out datasets, AFFINITYENHANCER outperforms SOTA structure-conditioned and inverse-folding baselines, with variants that improve lead-antibody binding without any antigen information (section 5).

## 2 BACKGROUND & RELATED WORK

**Structure-based design.** Most ML models targeted at antibody design, including the design of target-specific antibody libraries rely on structure-conditioned sequence generation, templated on the structure of the lead antibody or, when available, the structure of the antibody-antigen complex (Dreyer et al., 2023; Mahajan et al., 2022; Watson et al., 2023). Such structure-conditioning is necessary in order to restrain the designed sequences to adhere to the shape of the lead antibody. The sequence space can be further controlled when the structure of the antibody-antigen complex is known. For antibody design, in particular, structure-conditioning models such as AbMPNN (Dreyer et al., 2023), AntiFold (Høie et al., 2024), FvHallucinator (Mahajan et al., 2022) and MaskedProtEnT Mahajan et al. (2025) have demonstrated impressive performance on in silico benchmarks. On the other hand, de novo models such as RFDiffusion (Watson et al., 2023), follow a two-step process. First, they design the backbone of the antibody given the context of the antigen, then follow by sequence design with ProteinMPNN conditioned on that backbone in complex with the antigen.

**Sequence-based design.** Alternatively, sequence-only models have been proposed to generate protein or antibody sequences from a learned distribution or near the seed. Examples of such models include discrete Walk Jump Sampler Frey et al. (2023), latent Walk Jump Sampler, ProGen2 (Nijkamp et al., 2022), as well as language-model– and latent-space–guided directed evolution methods such as Hie et al. (2024); Tran & Hy (2024) and Tran et al. (2025). The latter demonstrate that large protein language models or latent generative models can effectively prioritize mutations during iterative directed evolution campaigns, improving protein function given repeated rounds of target-specific screening. However, there are no existing approaches which address affinity enhancement in a one-shot setting, and, in the absence of the antibody-antigen complex structure. Even de novo models such as RFDiffusion only guarantee binders (not improved binders) given a binding partner or antigen.

**Training with matched datasets.** We adopt a *matching-based supervision scheme* in which training pairs are formed by selecting, for each anchor the nearest neighbor such that (i) it lies within an input-space radius (ii) achieves a strictly higher measured affinity. This construction follows the spirit of PropEn which demonstrated that matching, implicitly recovers the ascent directions for a property of interest. Here we extend the matching to the one-shot antibody setting by including structure-aware embeddings and explicit environment control. In other words, in the requirements for matching, we add: (iii) pairing antibodies targeting the same antigen, i.e. *same environment*. Unlike PropEn, which uses sequence representation only, the AFFINITYENHANCERS matching operates in a geometry induced by pretrained encoders and a residual graph transformer to map low-affinity embeddings to higher-affinity counterparts.

Conceptually, this pairing induces *pairwise preferences* ($x' \succ x$), connecting our approach to preference learning (Zhang & Ranganath, 2025) methods such as Direct Preference Optimization (DPO) (Rafailov et al., 2023) from the LLM literature, where models are updated toward preference winners under KL regularization. Preference Learning has recently inspired a new direction in protein design. For backbone generation, Huguet et al. (2024) introduce Reinforced Fine-Tuning (ReFT): a supervised fine-tuning pass on a dataset filtered by auxiliary rewards to create a *preferential* subset, effectively supervised fine tuning on matched positives. In antibody co-design, Zhou et al. (2024) learn over *paired* samples by defining residue-level energy preferences and optimize a conditional diffusion model with a direct preference objective showing gains via energy decomposition and gradient-surgery to resolve conflicts. For peptide/protein binder design, (Mistani & Mysore, 2024)

explicitly formulate multi-objective alignment with DPO on curated chosen/rejected receptor-binder pairs, demonstrating that preference learning on matched datasets steers a protein LM toward binders satisfying specificity and developability (e.g., pI) constraints.

Although our setup is related to preference learning, there are two key differences. (1) Data and pairing. Preference learning typically uses (winner, loser) pairs indicating which sample has higher reward, without requiring the pair to be close in sequence space or to have similar measured values. In AFFINITYENHANCER, we construct local improvement pairs—nearby sequence variants that differ in measured binding—so that the learned transformation corresponds to realistic, stepwise improvements. (2) Optimization target. Preference learning usually updates a generator that maps a conditioning input (e.g., a receptor) to an output (e.g., a binder) to increase reward. In contrast, AFFINITYENHANCER performs lead-conditioned improvement: given an existing binder (the lead), it generates variants that are predicted to bind more strongly.

## 3    METHOD - AFFINITY ENHANCER

We formalize AFFINITYENHANCER as learning from matched improvements under fixed environments. In what follows, we state the data-generative model from which training pairs are drawn. Then, we derive the constraints that make the signal dominantly causal.

**Problem setup & method summary.** Let $\mathcal{X}$ denote the space of antibody sequences and let $\mathcal{Y} \subset \mathbb{R}$ denote measured binding affinities. We assume access to *E environments* indexed by $e = 1, \ldots, E$, where each environment corresponds to a distinct lead antibody (we use the terms leads or "seeds" interchangeably). In environment $e$ we observe a small subset of sequences (order of 10) with measured affinities $\{(x_j^e, y_j^e)\}$. Our goal is, for a held-out environment $e^*$ (the "one-shot" seed corresponding to an antigen not seen in the training set), to propose a set of new designs that reliably improve on the lead affinity $y_{\text{lead}}^{e^*}$, despite never fine-tuning on $e^*$ or using its antigen structure. To do so, we propose AFFINITYENHANCER, summarized in Figure 1:

1. **Form matched pairs.** $\mathcal{M} = (x_i, x_i' | e = e')$ in every environment $e$ by finding, for each low-affinity sequence $x_i$, the nearest neighbors $x_i'$ whose measured affinity is $y_i' > y_i$, under a capped distance threshold $\delta_x$.

2. **Extract embeddings.** For each antibody in the matched pairs, extract sequence-structure embeddings from a foundational model $\psi : \mathcal{X} \to \mathbb{R}^{L \times d}$.

3. **Learn a worse embedding→ better embedding map.** Given matched embeddings, learn a *Graph Transformer* $G_\theta$ acting over residues

$$f(z) := z + G_\theta(z; A, P), \qquad z = \psi(x) \in \mathbb{R}^{L \times d},$$

   where $A$ is a residue–residue adjacency (from predicted structure) and $P$ pos/edge features.

4. **Embeddings→sequence decoder.** Train a light-weight decoder $\rho : \mathbb{R}^{L \times d} \to \mathcal{X}$ that maps per-residue embeddings to amino-acid distributions.

5. **Sampling for OOD lead.** At test time, compute $z_{\text{lead}} = \psi(x_{\text{lead}}^{e^*})$ and apply residual map

$$\tilde{z} = f(z_{\text{lead}}) = z_{\text{lead}} + G_\theta(z_{\text{lead}}; A, P), \qquad \tilde{x} = \rho(\tilde{z}).$$

### 3.1    FROM MATCHED DATA TO CAUSAL SIGNALS

**Data-generative process.** We posit that $x$ factors into latent components:

$$x = f(s, c) \quad y = h(c, e)$$

where $c$ collects the **causal factors** that determine affinity and $s$ collects **spurious factors** (such as batch effects, library or lead/antigen idiosyncrasies, etc.) that influence $x$ but not $y$ once the environment $e$ is fixed. $c$ causally affects $y$ for fixed $e$. In an idealized world, we would sample independently

$$c \sim q(c), \quad s \sim q(s), \quad e \sim q(e), \quad x = f(s, c), \quad y = h(c, e).$$

In practice, *selection* (Pearl, 2009) induces dependencies: only some $(c, s)$ are assayed, and not every $x$ is tested in every $e$. The observable joint is therefore summarized as

$$(c, s) \sim p(c, s), x = f(s, c), \quad e \sim p(e \mid x), y = h(c, e),$$

In particular, $y$ depends on the target i.e. environment $e$, hence $s$ and $e$ may spuriously correlate with $y$ through selection rather than causation.

**Matched pair selection as targeted conditional.** For each anchor $x$ assayed in environment $e$ with outcome $y$, we seek a nearby variant $x'$ that improves the outcome, *in the same environment*:

$$p(x'|x, d(x, x') < \varepsilon, y' - y > \Delta y, e' = e) \tag{1}$$

with distance $d$ on $\mathcal{X}$, a small neighborhood radius $\varepsilon > 0$, and an improvement margin $\Delta y > 0$. Conditioning on $e' = e$ removes environment-driven gains; only changes in $x$ can explain improvements. This conditional represents the data-matching rule that defines our train set. For simplicity we include a deterministic analysis free of measurement noise.

**Assumption 1** (Property smoothness). *For fixed $e$, the property function is $K_y$-Lipschitz in the causal latent:*

$$|h(c_1, e) - h(c_2, e)| \leq K_y d(c_1, c_2).$$

**Assumption 2** (Responsive observation/bi-Lipschitz renderer). *There exists $K_x$ such that for all $(c, s), (c', s')$,*

$$\frac{1}{K_x} d([c, s], [c', s']) \leq d(f(s, c), f(s', c')) \leq K_x d([c, s], [c', s']),$$

*and the latent metric decomposes additively,*

$$d([c, s], [c', s']) = d(c, c') + d(s, s').$$

Intuitively, small moves in $x$, imply small moves in the underlying factors; no large cancellation can hide a big change in $c$ by counter-moving $s$.

**Theorem 1** (Improvement Bounds). *Consider a* matched *pair $(x, x')$ measured in the same environment with*

$$d(x, x') < \varepsilon \qquad and \qquad y' - y = h(c', e) - h(c, e) > \Delta_y > 0. \tag{2}$$

*Then:*

*1. (**Minimum causal movement**)*

$$d(c', c) > \Delta_y / K_y. \tag{3}$$

*2. (**Spurious-movement cap**) If, in addition, $K_x \varepsilon - \Delta_y / K_y \geq 0$, then*

$$d(s', s) < K_x \varepsilon - \Delta_y / K_y. \tag{4}$$

*Proof.* From equation 2 and A1,

$$\Delta_y < h(c', e) - h(c, e) \leq K_y d(c', c) \quad \Rightarrow \quad d(c', c) > \Delta_y / K_y,$$

which proves equation 3. Next, by equation 2 and A2,

$$d(c', c) + d(s', s) \leq K_x d(f(s, c), f(s', c')) \leq K_x \varepsilon.$$

Subtracting the lower bound on $d(c', c)$ from the left-hand side yields

$$d(s', s) < K_x \varepsilon - d(c', c) \leq K_x \varepsilon - \Delta_y / K_y,$$

establishing equation 4 whenever the right-hand side is nonnegative. □

The matching rule is feasible only if $K_x \varepsilon - \frac{\Delta y}{K_y} \geq 0$; otherwise no pair can simultaneously be close in $x$ and improve $y$. From equation 3 and equation 4, each pair enforces a minimal step along causal directions and leaves a strictly bounded budget for spurious drift. Hence, the supervision from matched improvements is dominated by *causal variation*.

**Training AFFINITYENHANCER** Let $z = \psi(x)$ be a sequence-structure embeddings (frozen). The embedding-to-embedding module learns a residual map $f_\theta(z) = z + G_\theta(z; A, P)$, trained to reconstruct matched targets in embedding space, by minimizing

$$\mathcal{L}(\theta) = \frac{1}{|M|} \Sigma_{(x,x') \in \mathcal{M}} ||\psi(x') - f_\theta(\psi(x))||_2^2.$$

At test time, for a held-out lead $x_{lead}$ in unseen environment $e^*$ we compute $z_{lead} = \psi(x_{lead})$, apply the residual map $\tilde{z} = f(z_{\text{lead}})$, and decode $x^* = \rho(\tilde{z})$.

*Why this objective isolates causal signals?* By equation 1 and equation 2, each training pair constrains the model with a guaranteed minimum shift in the causal coordinates and a tight upper bound on spurious motion. Averaged over many environments, spurious directions fluctuate and cancel, while causal directions align across pairs; minimizing $\mathcal{L}$ therefore compels $G_\theta$ to model the shared environment-invariant components that consistently explain affinity gains.

Given the selection rule equation 1 and the assumptions, every matched pair obeys

$$d(c', c) > \Delta y / K_y \quad and \quad d(s', s) < k_x \varepsilon - \Delta y / K_y,$$

so the training signal is *necessarily* a causal movement plus a bounded spurious residue. AFFINITYEN-HANCER exploits this by learning a residual embedding-space operator that reconstructs matched targets and, at inference steps in the same causal direction on held-out seeds. This "invariance-by-matching" view will underlie all experiments that follow.

## 4 AFFINITYENHANCER IMPLEMENTATION

Our theoretical formulation proposed above lends a direct implementation in our AFFINITYEN-HANCER which consists of three main modules ( Figure 1A). The structure and sequence embedder (Embedder), the reconstruction module and the embeddings to sequence decoder (Decoder) module. The Embedder embeds the antibody sequence and structure to a semantically meaningful embedded space. To this end, we utilize GearNet Zhang et al. (2023), a representation learning model trained on 600k sequences and structures from the AlphaFold2 database. To map the embeddings to antibody sequence, we trained a sequence decoder which maps GearNet (frozen) embeddings to antibody sequences on the paired Observed Antibody Space (pOAS), (Olsen et al., 2022). Once the sequence decoder is trained, it is also frozen. The reconstruction module, a Graph Transformer (GT), learns to reconstruct the embedding of the lower affinity antibody to the embedding of the higher affinity antibody. The reconstruction module is trained on the matched datasets prepared from SKEMPI 2.0. These modules allow us to embed sequences to a general embedding space that is trained on a massively large database of protein and antibody sequence and residue environments. Utilizing these pretrained modules allows us to leverage learned representations from all proteins and antibodies and generalize to blind or unseen test seeds.

## 5 EXPERIMENTS

The main challenge we address is whether it is possible to propose sequences of affinity enhanced designs starting from a single lead antibody sequence without *any* context or structure related to the antigen. Our validation pipeline is included in Figure 1B. We train AFFINITYENHANCER on a matched dataset that excludes any sequences in the vicinity of held-out seeds. Additionally, we utilize a predictive model, Cortex, (Gruver et al., 2023) (Appendix D) trained and validated on labeled expression and high-quality affinity data in vicinity of the held-out seeds. We then propose designs with AFFINITYENHANCER and predict the binding and affinity for the proposed designs with the oracle.

**Metrics.** We evaluate sampled designs by reporting edit distances from the seed sequence, the number of designs that are predicted to be binders, and number of improved binders over the seed. Additionally we include the binding and improved rates as well as the average performance across seeds to ease summarizing the performance per baseline.

**Baselines.** We compare AFFINITYENHANCER to three baselines – PropEn, trained on the same matched dataset as AFFINITYENHANCER, AntiFold, an antibody-specific, structure-conditioned

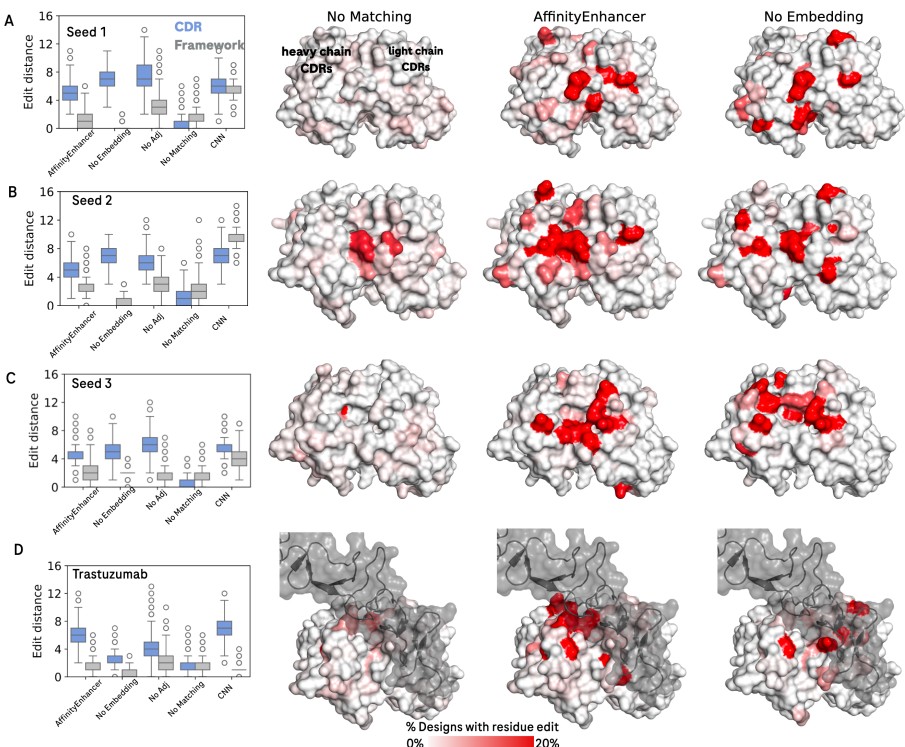

Figure 2: AFFINITYENHANCER identifies distinct and structurally important positions for each antibody. Each residue on the surface representation of the seed antibody is colored on a spectrum ranging from positions modified by the model in 0.0 (white) percent of designs to 20 percent (red) of designs. For Trastuzumab, we also show the antigen in gray. All antibody structure models were obtained with ESMFold. For Trastuzumab where the crystal structure of the antibody-antigen complex structure is available, we aligned the ESMFold structure to the crystal structure to map the position of the antigen.

inverse folding model and IgCraft (Greenig et al., 2025), an antibody-specific generative inpainting model. For further details on positioning AE wrt baselines, see Table 3.

**Ablations.** We systematically explore the effect of each component in AFFINITYENHANCER, dataset matching, embeddings, adjacency information and model architecture (local sequential kernels - convolutional neural networks, versus adjacency-informed graph transformers).

**Held-out seeds.** We evaluate in a true one-shot regime on four held-out seeds (three internal antibodies and the public Trastuzumab), each substantially out-of-distribution from training (full-sequence edit distance 64–87; Table 4). Table 5 quantifies potential germline overlap between train and test sets.

## 5.1 RESULTS

**AFFINITYENHANCER targets edits that retain and improve binding.** We asked how each model prior localizes affinity-enhancing edits across the antibody and, when known, at the antibody–antigen interface. In Figure 2A–D and Figure 9 we compare edit distance (leftmost panels) with the positions of edited residues on the binding surface (top-view CDRs). The model without matching ("No matching") serves as the baseline: it proposes few, nearly uniform edits across CDRs and frameworks, with no clear positional preferences (aside from Seed 2). In contrast, models trained with the matching intervention show distinct spatial patterns. The CNN variant makes more edits overall, spanning both CDRs and frameworks; Graph-Transformer (GT) variants concentrate edits in CDRs; and the "No Embedding" ablation makes the fewest framework edits. Across seeds, matched models repeatedly target protruding CDR motifs. For Trastuzumab, where the interface is known, many edited positions fall in direct contact with the antigen (Figure 2D).

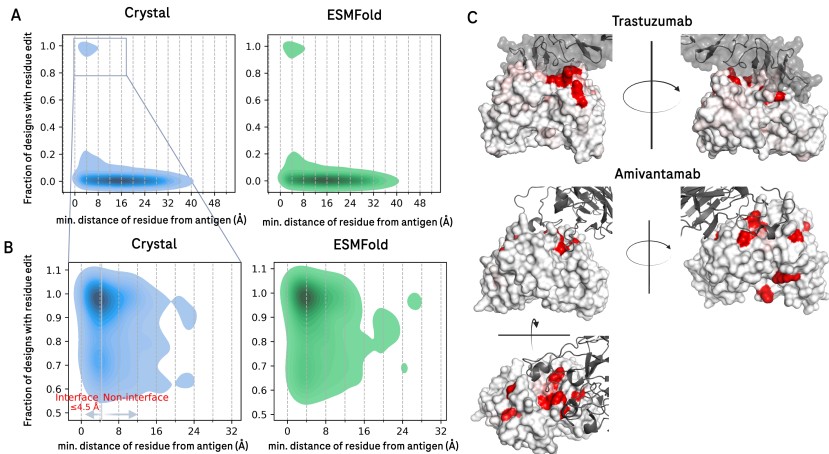

Figure 3: AFFINITYENHANCER primarily edits positions at the rim of the antibody-antigen interface. A) Kernel density plots for the distribution of frequency of residue edited by AE versus its distance from the antigen interface for a test set of 96 blind seeds with crystal or ESMFold structures of the starting seed. B) Same as A) but zoomed to show the density for the most edited positions C) (Top) Trastuzumab in complex with HER2 (PDB ID: 1N8Z) Cho et al. (2003). (Bottom) Amivantamab in complex with human MET protein(PDB ID: 6WVZ) Neijssen et al. (2021). Most edited positions by the AFFINITYENHANCER are colored red. ESMFold structures of the seeds were used for C) and edits were mapped back to the crystal structure of the complex. Proposed affinity-enhancing positions are concentrated in the rim as opposed to the core of the binding surface.

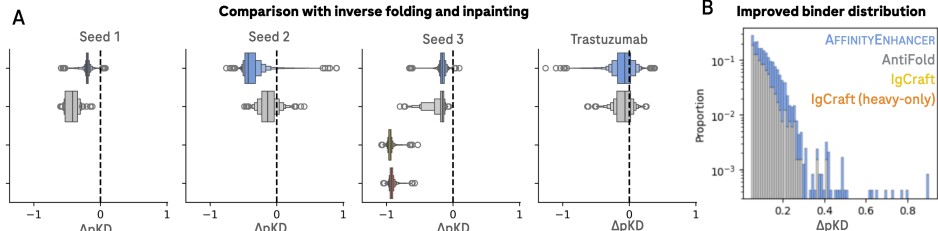

Figure 4: One-shot guided sampling with AFFINITYENHANCER. A) Comparison of AFFINITYEN-HANCER with the antibody-specific structure-conditioned inverse folding model, AntiFold and inpainting model IgCraft. Distribution of predicted $pKD$ (negative log10 of dissociation constant KD for unique designs with edit distance between [5,12] for 3 internal seeds and the Trastuzumab antibody. We report difference of the predicted $pKD$ from the $pKD$ of the seed. IgCraft designs were sampled for all-CDRs (IgCraft) or heavy chain CDRs only (IgCraft (heavy-only)). B) Distribution of affinity improvement for AFFINITYENHANCER, AntiFold and IgCraft for designs with improved affinities ($\Delta$ pKD$> 0.05$).

### 5.1.1 OUTPERFORMING INVERSE FOLDING AND INPAINTING BASELINES.

Figure 4 shows that AFFINITYENHANCER shifts the predicted-affinity distribution upward and yields more improved binders than baselines across all held-out seeds. Across all seeds, AFFINITYEN-HANCER shifts the predicted-affinity distribution decisively upward relative to AntiFold.Whereas AntiFold—by conforming to the seed antibody's structure—mostly proposes variants that retain binding with similar or lower affinity, AFFINITYENHANCER consistently produces affinity-improving designs for nearly every seed. The inpainting sequence based model IgCraft fails to propose CDR sequences (all CDRs or heavy-only CDRs) which retain or improve binding given the context of the framework residues. This further strengthens our claim that models which learn antibody sequence distributions are insufficient to generate CDR sequences that retain binding. The magnitude of these gains also exceeds those from both AntiFold and the inpainting model IgCraft (Figure 3B).

Table 1: **Ablation study for AFFINITYENHANCER.** We sample 5,000 sequences for Trastuzumab and for three internal seeds per model. ED = minimum edit distance to the parent; "ED window" counts designs with ED $\in [5, 12]$. "Binders" and "Improved binders" are predicted positives and affinity-improved positives, respectively. AFFINITYENHANCER uses GearNet embeddings, pOAS decoder, an adjacency-informed Graph Transformer, and data matching. PropEn is the sequence-only baseline from Tagasovska et al. (2024). For each model, sampling settings were chosen to maximize the number of designs sampled in the ED $\in [5, 12]$ range.

| Model | Set | ED | ED window | Binders | Improved | Binder rate | Improved rate |
|---|---|---|---|---|---|---|---|
| AFFINITYENHANCER | Trastuzumab | $7.9 \pm 1.8$ | 4,815 | **3,970** | 1,575 | **79.4** % | 31.50 % |
| | Seed 1 | $6.5 \pm 1.6$ | 4,382 | **1,105** | 2 | 22.1 % | 0.04 % |
| | Seed 2 | $7.4 \pm 1.8$ | 4,672 | **3,612** | 113 | 72.2 % | 2.26 % |
| | Seed 3 | $6.5 \pm 1.7$ | 4,352 | **1,334** | 3 | 26.7 % | 0.06 % |
| | *Mean over seeds* | 7.08 | 4,555 | 2,505 | 423 | **50.10** % | 8.46 % |
| | *Seeds improved (Trastuzumab + Seeds 1–3)* | | | | | **4** | **/4** |
| PropEn (– *Structure*) | Trastuzumab | 28.6 | 0 | 0 | 0 | 0.0 % | 0.00 % |
| | Seed 1 | 68.0 | 0 | 0 | 0 | 0.0 % | 0.00 % |
| | Seed 2 | 30.9 | 0 | 0 | 0 | 0.0 % | 0.00 % |
| | Seed 3 | 68.4 | 0 | 0 | 0 | 0.0 % | 0.00 % |
| | *Mean over seeds* | 55.8 | 0 | 0 | 0 | 0.0 % | 0.00 % |
| | *Seeds improved (Trastuzumab + Seeds 1–3)* | | | | | **0** | **/4** |
| AFFINITYENHANCER (– *Matching*) | Trastuzumab | 2.8 | 447 | 392 | 162 | 7.8 % | 3.24 % |
| | Seed 1 | 2.6 | 253 | 45 | 0 | 0.9 % | 0.00 % |
| | Seed 2 | 3.4 | 954 | 838 | 696 | 16.8 % | **13.92** % |
| | Seed 3 | 2.3 | 98 | 47 | 0 | 0.9 % | 0.00 % |
| | *Mean over seeds* | 2.78 | 438 | 331 | 215 | 6.61 % | 4.29 % |
| | *Seeds improved (Trastuzumab + Seeds 1–3)* | | | | | **2** | **/4** |
| AFFINITYENHANCER (– *Embedding*) | Trastuzumab | 3.1 | 161 | 39 | 14 | 0.8 % | 0.28 % |
| | Seed 1 | 7.1 | 2,366 | 171 | 134 | 3.4 % | **2.68** % |
| | Seed 2 | 7.6 | 3,486 | **3,457** | 112 | 69.1 % | 2.24 % |
| | Seed 3 | 7.3 | 1,992 | 1,737 | 4 | 34.7 % | 0.08 % |
| | *Mean over seeds* | 6.27 | 2,001 | 1,351 | 66 | 27.02 % | 1.32 % |
| | *Seeds improved (Trastuzumab + Seeds 1–3)* | | | | | **4** | **/4** |
| AFFINITYENHANCER (– *Graph Transformer*) | Trastuzumab | 8.0 | 4,570 | 2,027 | 124 | 40.5 % | 2.48 % |
| | Seed 1 | 11.2 | 3,812 | 1,085 | 0 | 21.7 % | 0.00 % |
| | Seed 2 | 17.0 | 13 | 13 | 0 | 0.3 % | 0.00 % |
| | Seed 3 | 9.4 | 4,719 | 89 | 2 | 1.8 % | 0.04 % |
| | *Mean over seeds* | 11.40 | 3,279 | 804 | 32 | 16.07 % | 0.63 % |
| | *Seeds improved (Trastuzumab + Seeds 1–3)* | | | | | **2** | **/4** |
| AFFINITYENHANCER (– *Adjacency Matrix*) | Trastuzumab | 6.8 | 4,196 | 3,297 | **1,951** | 65.9 % | **39.02** % |
| | Seed 1 | 10.7 | 3,939 | 179 | 8 | 3.6 % | 0.16 % |
| | Seed 2 | 9.1 | 4,769 | 2,873 | 38 | 57.5 % | 0.76 % |
| | Seed 3 | 7.2 | 4,423 | 659 | 0 | 13.2 % | 0.00 % |
| | *Mean over seeds* | 8.45 | 4,332 | 1,752 | 499 | 35.04 % | **9.98** % |
| | *Seeds improved (Trastuzumab + Seeds 1–3)* | | | | | **3** | **/4** |

### 5.1.2 ABLATIONS: WHICH COMPONENTS OF AFFINITYENHANCER MATTER AND WHY.

*Sequence-Only Baseline*: We first compare AFFINITYENHANCER to PropEn (sequence-only) across Seed 1–3 and Trastuzumab (*Table 1*). PropEn proposes designs more than 25 edits from the seed in every case, i.e., it fails to generate variants in the seed's neighborhood; none of its designs are predicted binders.

Across all seeds, AFFINITYENHANCER generates designs close to the seed (Tables 1), with 26–78% predicted binders and non-zero counts of improved binders for each seed. Edit distance is controllable via sampling (iterations/temperature), enabling small-to-moderate edits at low settings and larger edits at higher settings (Tables 2–5, Figure 8).

*(– Matching) Autoencoder Without Guidance*: Removing the matching intervention reduces the model to an embedding-space autoencoder. This yields low-diversity proposals clustered near the seed and few binders or improved binders (notable exception: Seed 2). Matching is therefore critical for shifting probability mass toward functional, higher-affinity regions.

*(– Embedding) Generalization Without the Embedder*: Without GearNet embeddings and the pOAS decoder, the model still produces some improved binders across all seeds and, for Seeds 1 and 3, the most improved binders among ablations. This suggests that structural priors plus matching capture useful causal signal even without the embedder. However, sequence diversity and binder counts—especially for Trastuzumab—lag the full model. Furthermore, edit distances are less controllable and limited to a single iteration (Tables 2–5, Figure 8).

*(CNN)  Weaker Structural Prior, Weaker Binders*: Replacing the Graph Transformer with a CNN (PropEn-style) increases edit distances and weakens edit-distance control (Tables 2–5, Figure 8). Binder and improved-binder yields drop substantially, indicating that the GT's relational bias is important for localized, functional edits.

*(– Adjacency)  Losing Contacts, Losing Control*: Using a fully connected Graph Transformer (no adjacency) similarly inflates edits and reduces controllability with sampling knobs (Tables 2–5, Figure 8). This highlights the role of explicit adjacency in guiding compact, physics-aware modifications.

### 5.1.3 COMPARISON TO EXPERIMENTAL DATA AND BIOLOGICAL INSIGHTS

We probe what AFFINITYENHANCER (AE) learns beyond improving an in silico oracle score: (i) whether its edits localize to biologically plausible regions of the binding interface without antigen context, and (ii) whether the positions and substitutions it prioritizes align with experimentally validated affinity effects. We focus on publically available, structurally characterized, out-of-distribution (OOD) antibody seeds with low similarity to training and no germline overlap (details in Appendix E.2).

**AE localizes edits to the interface rim without complex structures.** Mapping per-residue edit frequency onto solved antibody–antigen complexes shows a consistent pattern: AE concentrates edits near the *rim* of the interface, while making few edits in the *core* (Figure 3) and non-interface residues. This is biologically meaningful because affinity improvements, particularly from already, strong leads—often arise by extending or refining peripheral contacts rather than perturbing core interactions. The same trend holds within CDRs: the most frequently edited CDR residues are enriched at the interface rim (Figure 5). Together, these results suggest AE infers binding-relevant regions from antibody-only input, without using antigen information at train or test time.

**AE highlights experimentally high-impact positions.** For the G6 antibody, where large-scale single-mutant affinity measurements are available Koenig et al. (2017), positions most frequently edited by AE are associated with larger experimentally observed improvements; in particular, the top edited positions include those attaining the maximum measured gain (Figure 6A). On three additional seeds from Minot & Reddy (2024) with thousands of experimentally assayed variants (bind/no-bind), AE's top-edited positions avoid sites where substitution almost always abolishes binding and instead concentrate on positions exhibiting informative, context-dependent effects (Figure 6B). This behavior is consistent with learning where edits are both impactful and tolerable.

**AE proposes semantically meaningful substitutions.** In cases where experiments indicate electrostatics-driven improvements, AE proposes qualitatively consistent substitutions. For Seed 1, experimentally beneficial mutations include substitutions to negatively charged residues (D/E); AE similarly proposes such substitutions and yields variants with correspondingly more negative electrostatic surface at the relevant CDR region (Figure 7A).

## 6 CONCLUSION

In this work, we tackle the one-shot task of affinity maturing a lead antibody for blind or unseen seeds. AFFINITYENHANCER combines dataset matching with pretrained sequence–structure representations, an antibody-specific decoder, and lightweight structural priors to propose targeted edits directly from the lead sequence. Empirically, we show it recovers binding-relevant features from sequence alone and generates affinity-enhancing mutations. Across held-out evaluations, it outperforms sequence-only PropEn, a structure-conditioned inverse-folding baseline, and a sequence-inpainting model, sampling variants with consistently higher affinity. Unlike reconstruction-driven approaches, AFFINITYENHANCER is designed to discover *causal*, affinity-improving mutations—yielding practical gains for directed evolution.

Beyond accuracy, AFFINITYENHANCER offers practical advantages for directed evolution: it generalizes to new seeds in a one-shot regime, provides controllable edit distances for risk-aware exploration, and remains data-efficient by leveraging pretrained biomolecular priors. These properties make it a useful drop-in companion for antibody lead optimization when structural complexes or large labeled datasets are unavailable. Looking ahead, AFFINITYENHANCER creates a clear path for further gains. Incorporating epitope or antigen context could disambiguate multiple plausible routes to improvement, while expanding labeled affinity resources will broaden coverage of binding modes.

## 7 REPRODUCIBILITY STATEMENT

We will release (i) the exact SKEMPI-derived matched pairs (IDs and thresholds), (ii) code to recompute matches from raw SKEMPI/pOAS, (iii) all hyperparameters, (iv) pre-trained weights for  and $G_\theta$, and (v) scripts to reproduce all figures/tables from a single make entrypoint. We report complete sampling settings and will upload an anonymous artifact with code and models at submission time.

## ACKNOWLEDGMENTS

We thank Jan Ludwiczak for expertly curating and quality-controlling antibody–antigen complex structures from the SAbDab dataset used in this work.

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

APPENDIX

CONTENTS

## A  PROBABILISTIC BOUNDS FOR THEOREM 1

We now allow *only* affinity measurements to be noisy, with noise *independent* of the environment. The sequence/embedding path is noise-free. Specifically,

$$z = \psi(x), \qquad y_{\text{obs}} = h(c, e) + \xi_y,$$

where $\xi_y$ is zero-mean sub-Gaussian with proxy $\sigma_y^2$, i.i.d. across samples, and independent of $(e, c, s)$. (The sub-Gaussian assumption yields tight, environment-agnostic high-probability margins; weaker moment assumptions are possible with looser bounds.)

**Observed matching rule.** We form matches using *observed* improvements in the same environment and exact embedding proximity:

$$d\big(\psi(x), \psi(x')\big) < \varepsilon, \qquad y'_{\text{obs}} - y_{\text{obs}} > \Delta_y, \qquad e = e'. \tag{5}$$

**High-probability causal movement.** Since $y'_{\text{obs}} - y_{\text{obs}} = \big(h(c', e) - h(c, e)\big) + (\xi'_y - \xi_y)$, sub-Gaussianity implies that for any $\delta \in (0, 1)$ there exists

$$\Gamma_y(\delta) = 2\sigma_y\sqrt{2\log(1/\delta)} \quad \text{s.t.} \quad \mathbb{P}(h(c', e) - h(c, e) > \Delta_y - \Gamma_y(\delta)) \geq 1 - \delta.$$

By A1, with the same probability,

$$d(c', c) \; > \; \frac{\Delta_y - \Gamma_y(\delta)}{K_y}. \tag{6}$$

**High-probability spurious cap (no $x$-noise).** Assume A2 and denote $z = \psi(x)$. From equation 5, $d(c', c) + d(s', s) \leq K_x \varepsilon$. Combining with equation 6 yields, with probability $\geq 1 - \delta$,

$$d(s', s) \; < \; K_x \varepsilon \; - \; \frac{\Delta_y - \Gamma_y(\delta)}{K_y}. \tag{7}$$

**Feasibility and interpretation.** Non-vacuous guarantees require $\Delta_y > \Gamma_y(\delta)$ and $K_x\varepsilon - (\Delta_y - \Gamma_y(\delta))/K_y \geq 0$. Because $\xi_y$ is independent of the environment, the margin $\Gamma_y(\delta)$ is *uniform across all $e$*. Equations equation 6–equation 7 are the noise-robust analogues of the deterministic bounds equation 3–equation 4 when $y$ is noisy.

## B  AFFINITY ENHANCER - MODEL AND TRAINING

Dataset: The matched dataset was prepared with an edit distance threshold of 5 and a $pKD$ threshold of 1.5.

**Inputs and embeddings**: All structures are predicted with ESMFold. Per-residue GearNet embeddings concatenated over all 6 layers (dimension=3072) are obtained for the full fv (heavy and light chains), followed by padding to a fixed heavy and light chain length of 151 and 150 respectively.

**Sequence decoder**: The sequence decoder is a 2-layer multi-layer perceptron with a hidden dimension of 32 and ReLU activation. The decoder is trained with the GearNet embeddings (frozen) on the paired Observed Antibody Space (pOAS) Table

**Model**: AFFINITYENHANCER has 4.2M parameters 2. The GraphTransformer was adapted from lucidrains implementation on Github (https://github.com/lucidrains/graph-transformer-pytorch). It has 4 blocks. Each block consists of normalization layer, an attention layer, a gated residual connection with 4 attention heads and a hidden dimension of 256. The model was trained for 200 epochs. Training times as a function of dataset size are reported in Table 2. Seeds 1, 2 and 3 were trained on trainset of size 2200 whereas Trastuzumab was trained on trainset of size 1300 (after removing all sequences in the vicinity of the seed). At inference, it takes 3-3.5 mins to generate 5000 samples (batched inference with batch size of 64) on a single A100 or a G5 GPU.

## C  AFFINITY ORACLE

For our in silico validation, we use Cortex (Gruver et al., 2023), a multi-task fine-tuning framework that leverages pre-trained language models for Biological Sequence Transformation and Evolutionary Representation (LBSTER) (Frey et al., 2024) to predict multiple properties of interest (e.g., binding and expression). Cortex has been trained on a diverse set of targets, including the leads and surrounding data used in this manuscript. On our in-distribution evaluation, Cortex achieves 82.9% mean binder-classification accuracy with 0.4% standard error, and a Spearman correlation of 0.90 with measured pKD. Per lead, binder accuracy is 72.4% (Seed 1), 62.0% (Seed 2), 70.0% (Seed 3), and 78.4% (Trastuzumab). This oracle has also been recently suggested in an extensive lab-in-the-loop study for antibody affinity maturation (Frey et al., 2025).

Table 2: Model size and training cost (A100).

| Model components | | | |
|---|---|---|---|
| **Component** | **Type** | **Params** | **Mode** |
| Embedder | GearNet | 20.1M | frozen |
| Decoder | MLP | 99.0K | train |
| Autoencoder (total) | GearNet_MLP | 20.2M | eval |
| Graph model | GraphTransformer | 4.2M | train |
| **Training time vs. matched dataset size** | | | |
| **#Pairs** | **Hours** | | **GPUs** |
| 1300 | ∼2.5 | | 1 |
| 2200 | ∼5 | | 1 |
| 5100 | ∼14 | | 1 |
| 7640 | ∼48 | | 2 |

Table 3: Positioning of AFFINITYENHANCER with respect to SOTA methods.

| | IID optimization | OOD optimization | single-shot | improved binders with CDR edits |
|---|---|---|---|---|
| **AFFINITYENHANCER (ours)** | ✓ | ✓ | ✓ | ✓ |
| **Property Enhancer (Tagasovska et al. (2024))** | ✓ | ✗ | ✗ | ✓ |
| **AntiFold ((Høie et al., 2024))** | ✓ | ✓ | ✗ | ✗ |
| **Walk-Jump, diffusion (Frey et al. (2023))** | ✓ | ✗ | ✓ | ✗ |
| **EffEVO, LM-based (Hie et al. (2024))** | ✓ | ✓ | ✗ | ✗ |
| **IgCraft (Greenig et al. (2025))** | ✓ | ✓ | ✓ | ✗ |
| **Directed Evolution** (Tran et al. (2025); Tran & Hy (2024)) | ✓ | ✗ | ✓ | ✗ |

# D    ANTIFOLD AND IGCRAFT

For AntiFold, for each seed, we sampled 5000 sequences at temperatures 0.2 and 0.5. For IgCraft, we sampled sequences with default parameters and for additional setting (lower sampling temperature of 0.05 and number of steps set to 10).

# E    ADDITIONAL RESULTS FROM EXPERIMENTS

## E.1    OVERLAP BETWEEN TRAINING SET AND TEST SEEDS

Table 4: Edit Distance and Sequence identity (SI) to trainset for test set seeds. Seeds 1-3 and Trastuzumab are used for affinity enhancement experiments. Amivantamab was used for structural comparison in Figure 3B. For reference, a typical heavy chain is 120 residues whereas a typical light chain is 106 residues in length. A sequence identity of 90-95% ( 10-30 residues) is commonly used to demarcate out-of-distribution samples for antibody sequences in prior works.

| Seed | SI | heavy | light | full | L1 | L2 | L3 | H1 | H2 | H3 |
|---|---|---|---|---|---|---|---|---|---|---|
| Seed1 | 63 | 42 | 37 | 84 | 4 | 2 | 5 | 5 | 6 | 8 |
| Seed2 | 69 | 28 | 35 | 69 | 3 | 3 | 3 | 2 | 4 | 6 |
| Seed3 | 71 | 30 | 18 | 64 | 4 | 2 | 4 | 4 | 5 | 4 |
| Trastuzumab | 72 | 28 | 33 | 64 | 6 | 3 | 5 | 3 | 5 | 6 |
| Amivantamab (structure comparison) | 68 | 44 | 19 | 73 | 3 | 3 | 2 | 6 | 5 | 5 |

## E.2    LEARNED BIOLOGICAL FEATURES

**OOD seed selection.** We selected 96 antibody seeds with experimentally solved antibody–antigen complex structures from SAbDab Dunbar et al. (2014). Antibodies were selected with the following criteria: 1) deposited since 2024 2) antigen type is protein 3) conventional antibody 4) resolution of the complex structure $\leq 3.0$ Å 5) sequence identity to the closest sequence in trainset $\leq 0.70$ and 6) No v-gene overlap with the trainset. We report similarity statistics of the structural dataset in Table 6.

Table 5: Overlap between train set and test set germlines. Seeds 1-3 and Trastuzumab are used for affinity enhancement experiments. Amivantamab was used for structural comparison in Figure 3B.

| | Matching gene in train set | | | |
|---|---|---|---|---|
| Seed | # heavy V-gene | # heavy J-gene | # light V-gene | # light J-gene |
| Seed 1 | 0 | 193 | 0 | 0 |
| Seed 2 | 0 | 0 | 0 | 1785 |
| Seed 3 | 0 | 85 | 0 | 179 |
| Trastuzumab | 0 | 72 | 0 | 813 |
| Amivantamab (Structure comparison) | 371 | 85 | 0 | 279 |

Table 6: Independent structural dataset with 96 antibody seeds with known antibody-antigen complexes and the maximum and minimum sequence identity (SI) and edit distances of the test set from the training set.

| Closest/Farthest | SI full | ED full | ED heavy | ED light |
|---|---|---|---|---|
| Closest | 0.70 | 69 | 13 | 19 |
| Median | 0.64 | 82 | 39.5 | 32.0 |
| Farthest | 0.52 | 103 | 63 | 57 |

**Sampling and structures used at inference.** For each seed, we generated designs with AE using antibody-only input. For comparison with experimentally measured affinities from (Minot & Reddy, 2024) and Koenig et al. (2017), we used the experimentally determined unbound antibody structure; otherwise, we used a predicted antibody structure from ESMFold. For results reported in Figure 3, we used both experimentally determined holo antibody structures and ESMFold (unbound) structures. For simplicity we sampled 2000 sequences per seed (temperature= 1.0 and iterations= 7 using the same decoding settings as in the main experiments.

**Edit-frequency computation.** For each seed, we computed per-position edit incidence as the fraction of sampled designs whose amino acid at that position differs from the seed. We optionally report the top-$k$ most frequently edited positions per seed.

**Interface distance and core/rim definition.** Using the antibody–antigen complex structure, we computed for each antibody residue the minimum heavy-atom distance to any antigen atom. We grouped residues by distance bins and plotted edit incidence as a function of distance (Figure 3). We define *interface* using distance thresholds (4.5 Åbetween any atom on an antibody residue and the closest atom on the antigen) and repeat the analysis restricted to CDR residues (Figure 5).

**Comparison to single-mutant affinity measurements (G6).** For the G6 antibody, we used published experimental measurements of affinity changes for single substitutions (over 4000 mutants). For each position, we computed the maximum observed affinity improvement across tested substitutions and compared this statistic between positions preferentially edited by AE and those not preferentially edited (Figure 6A). We define "preferentially edited" as the top 10th percentile of positions by edit incidence.

**Comparison to bind/no-bind variant datasets.** For three additional seeds with large-scale bind/no-bind measurements (thousands of variants each), we quantified for each position the empirical change in binding when that position is mutated relative to the seed, along with the variance of the binding outcome across substitutions. We then examined where AE's top-edited positions fall in this (mean-change, variance) space, verifying that AE avoids positions with near-deterministic loss of binding (high mean-change, zero variance) while emphasizing positions with context-dependent effects (Figure 6B).

**Electrostatic surface analysis.** For Seed 1, we compared electrostatic surface potentials for the seed, an AE-designed variant containing a negatively charged substitution in CDR H3, and a top experimentally validated single-point mutation with a D substitution in CDR H3. Electrostatic surfaces were computed using APBS plugin in PyMOL Lerner & Carlson (2006); Baker et al. (2001); Dolinsky et al. (2004), using the same structural backbone and standard protonation settings (Figure 7A).

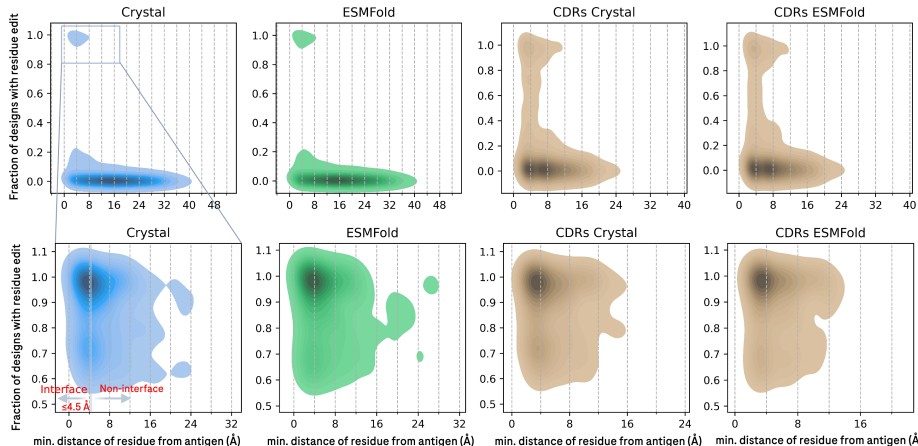

Figure 5: AFFINITYENHANCER primarily edits positions at the rim of the antibody-antigen interface. A) Kernel density plots for the distribution of frequency of residue edited by AE versus its distance from the antigen interface (minimum distance of any atom on antibody residue to antigen) for a test set of 96 blind seeds with crystal and ESMFold structures of the starting seed for all residues and cdrs only. B) Same as A) but zoomed to show the density for the most edited positions.

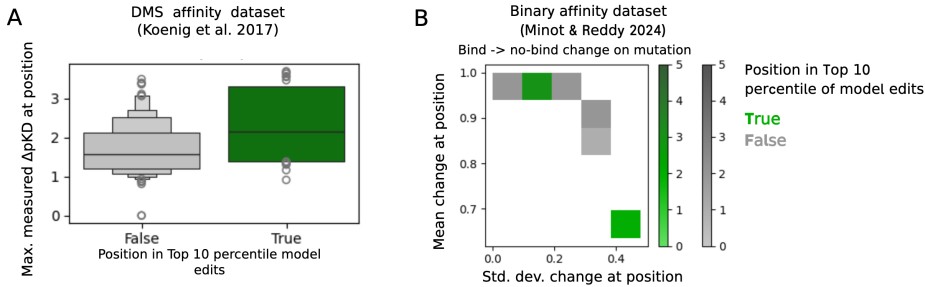

Figure 6: Comparison of AFFINITYENHANCER edits to experiments. A) Comparison to deep mutational scanning data from Koenig et al. (2017). Distribution of maximum experimental improvement in pKD over seed for a position edited by AFFINITYENHANCER. B) Comparison to bin/no-bind experimentally measured datasets for 3 seeds from Minot & Reddy (2024). Distribution of the mean versus the standard deviation of the change in measured binding from seed (i.e. bind to no-bind). Higher mean indicates important of the position to binding. High mean and zero standard deviation indicates a position which upon mutation with respect to the seed amino acid will always lead to loss of binding. High mean and high standard deviation implies a position important to binding but also available for edits. "True" refers to a position being present in the top 10 percentile of the positions edited by the model for that seed.

## F   COMPARISON OF AFFINITYENHANCER WITH ORACLE-GUIDED LATENT MODELS

A common class of protein optimization algorithms relies on generative models guided by oracles/predictors in latent space, with directed evolution approaches such as Tran & Hy (2024) and Tran et al. (2025) being representative examples. To compare AffinityEnhancer to this framework, we focus on Tran & Hy (2024) and use the authors' implementation as shared in the public repository. To adapt the method for binding affinity, we first train the ESM2-based decoder on the SKEMPI v2 dataset (after removing sequences close to Seed 1 and Trastuzumab for the one-shot setup). We use the default parameters dec hidden dim = 1280, batch size=256, lr=5e-5 , and num epochs=50. The validation MAE for the decoder is 0.451 for Seed 1 and 1.167 for Trastuzumab. We then sample new sequences with n steps=10, population=5000, num proposes per var=4 , population ratio per

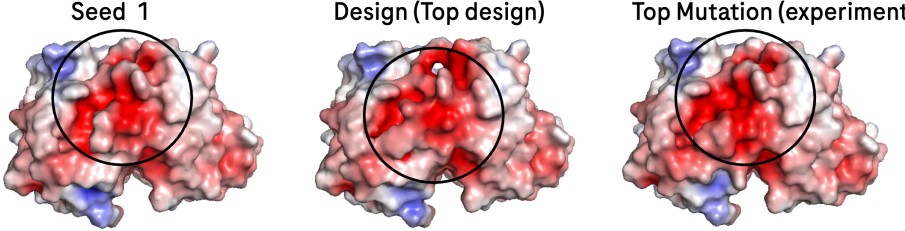

Figure 7: Comparison of amino acid substitutions with experiments. Comparison of the electrostatic surface of the Seed, top design and a top experimental single-point mutant for Seed 1. Regions with the amino acid substitution in question is circled. Color scale: Blue-positive $+5$; white-neutral; red-negative $-5$.

mask=0.6. We choose a lower number of steps than the default (60), since this yields designs closer to the seed, for which we have greater confidence in the oracle predictions.

We were unable to generate a sufficiently large number of designs for MLDETran & Hy (2024) with reasonable edit distances from the seed. For example, default settings yielded edit distances of >30 edits in the seed (Seed 1). We selected a lower number of iterations to obtain MLDE designs in the vicinity of the seed. To match this setting, we also generated AffinityEnhancer designs with lower temperatures and iterations. We also compared predicted affinity values between MLDE and AE and found later to give better affinities. AE also primarily identified edits in the CDRs versus MLDE which made edits in both CDRs and frameworks regions to the same extent. In all settings tested, AffinityEnhancer outperforms MLDE. (Table 7).

Table 7: Comparison of AffinityEnhancer (AE) for Seeds 1 and Trastuzumab with MLDETran & Hy (2024).

| Method | seed | ED | ED window | Binders | Improved | Binder rate | Improved rate |
|---|---|---|---|---|---|---|---|
| MLDE (low ED) | Seed 1 | $5.9 \pm 0.8$ | 98/128 | 32/98 | 0 | 34.7% | 0% |
| AE (low ED) | Seed 1 | $5.2 \pm 0.46$ | 283/497 | 103/283 | 0 | 36.4% | 0.0% |
| MLDE | Seed 1 | $16.2 \pm 1.0$ | 5/5000 | 0/5 | 0 | 0% | 0% |
| MLDE | Trastuzumab | $13.6 \pm 1.1$ | 816/5000 | 0/816 | 0 | 0% | 0% |
| AE | Seed 1 | $6.5 \pm 1.6$ | 4382/5000 | 1,105 | 2 | 22.1% | 0.04% |
| AE | Trastuzumab | $7.9 \pm 1.8$ | 4815/5000 | 3970 | 1575 | 79.4% | 31.5% |

### F.1 MODEL ABLATIONS

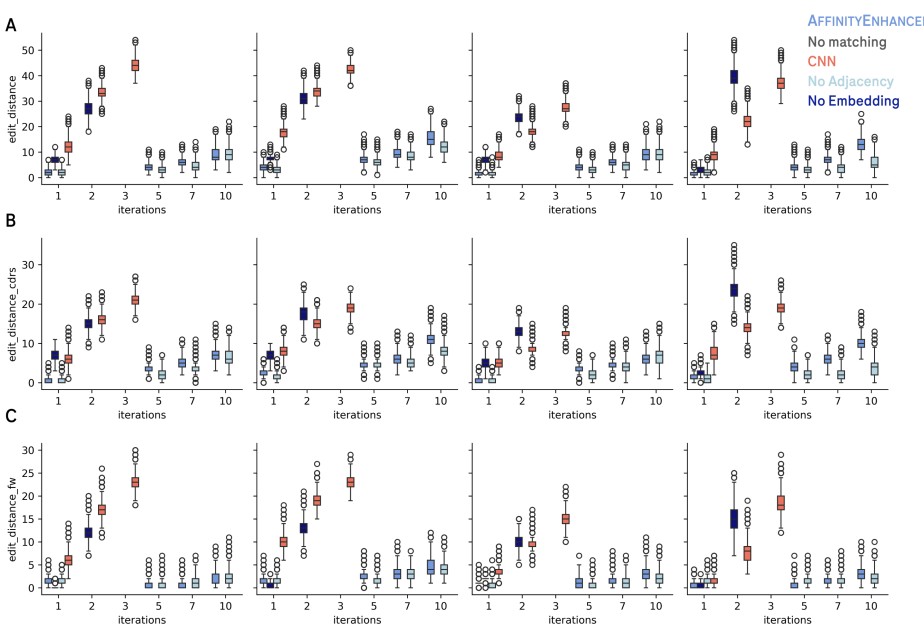

Figure 8: Edit distance distribution as a function of number of iterations at sampling for different model ablations A) Full length antibody B) CDRs only C) Framework regions only.

Table 8: Selecting sampling parameters for Trastuzumab with the maximum number of designs within an edit distance of [5,12] from the seed sequence. We sampled 5000 designs for 3 internal seeds and Trastuzumab. AFFINITYENHANCER is the base model with a GearNet embedder and pOAS sequence decoder, an adjacency-informed Graph Transformer, and data matching. PropEn is the sequence-only model from Tagasovska et al. (2024).

| Model | Iterations | Temperature | Edit distance | # in [5,12] |
|---|---|---|---|---|
| AFFINITYENHANCER | 1 | 0.7 | $1.1 \pm 0.3$ | 0 |
| | 1 | 1.0 | $2.0 \pm 0.9$ | 20 |
| | 5 | 0.7 | $3.4 \pm 0.9$ | 118 |
| | 5 | 1.0 | $4.8 \pm 1.5$ | 2544 |
| | 7 | 0.7 | $5.7 \pm 1.1$ | 2427 |
| | 7 | 1.0 | $7.9 \pm 1.8$ | 4815 |
| | 10 | 0.7 | $11.3 \pm 1.3$ | 3543 |
| | 10 | 1.0 | $14.6 \pm 2.2$ | 805 |
| AFFINITYENHANCER (No Matching) | 1 | 0.7 | $1.3 \pm 0.5$ | 0 |
| | 1 | 1.0 | $2.8 \pm 1.3$ | 393 |
| | 5 | 0.7 | $1.4 \pm 0.6$ | 0 |
| | 5 | 1.0 | $2.7 \pm 1.3$ | 361 |
| | 7 | 0.7 | $1.4 \pm 0.6$ | 0 |
| | 7 | 1.0 | $2.8 \pm 1.3$ | 407 |
| | 10 | 0.7 | $1.4 \pm 0.6$ | 0 |
| | 10 | 1.0 | $2.8 \pm 1.3$ | 447 |
| AFFINITYENHANCER (No Embed) | 1 | 0.7 | $2.5 \pm 0.9$ | 2 |
| | 1 | 1.0 | $3.1 \pm 1.1$ | 161 |
| | 2 | 0.7 | $37.2 \pm 2.9$ | 0 |
| | 2 | 1.0 | $41.0 \pm 3.5$ | 0 |
| AFFINITYENHANCER (CNN) | 1 | 0.5 | $7.3 \pm 1.3$ | 2319 |
| | 1 | 0.7 | $8.0 \pm 1.5$ | 4570 |
| | 1 | 1.0 | $10.8 \pm 2.1$ | 3988 |
| | 2 | 0.5 | $20.0 \pm 1.7$ | 0 |
| | 2 | 0.7 | $21.4 \pm 2.1$ | 0 |
| | 2 | 1.0 | $25.4 \pm 2.7$ | 0 |
| | 3 | 0.5 | $35.3 \pm 1.9$ | 0 |
| | 3 | 0.7 | $36.7 \pm 2.3$ | 0 |
| | 3 | 1.0 | $40.1 \pm 2.8$ | 0 |
| AFFINITYENHANCER (No Adj) | 1 | 0.7 | $1.2 \pm 0.4$ | 0 |
| | 1 | 1.0 | $2.5 \pm 1.2$ | 266 |
| | 5 | 0.7 | $1.6 \pm 0.7$ | 0 |
| | 5 | 1.0 | $3.5 \pm 1.6$ | 1049 |
| | 7 | 0.7 | $2.1 \pm 0.8$ | 6 |
| | 7 | 1.0 | $4.5 \pm 1.9$ | 2269 |
| | 10 | 0.7 | $3.2 \pm 1.0$ | 238 |
| | 10 | 1.0 | $6.8 \pm 2.2$ | 4196 |

Table 9: Selecting sampling parameters for Seed 1 with the maximum number of designs within an edit distance of [5,12] from the seed sequence. We sampled 5000 designs for 3 internal seeds and Trastuzumab. AFFINITYENHANCER is the base model with a GearNet embedder and pOAS sequence decoder, an adjacency-informed Graph Transformer, and data matching. PropEn is the sequence-only model from Tagasovska et al. (2024).

| Model | Iterations | Temperature | Edit distance | # in [5,12] |
|---|---|---|---|---|
| AFFINITYENHANCER | 1 | 0.7 | $1.1 \pm 0.3$ | 0 |
| | 1 | 1.0 | $2.2 \pm 1.0$ | 83 |
| | 5 | 0.7 | $3.3 \pm 0.7$ | 33 |
| | 5 | 1.0 | $4.6 \pm 1.3$ | 1985 |
| | 7 | 0.7 | $4.8 \pm 0.9$ | 1148 |
| | 7 | 1.0 | $6.5 \pm 1.6$ | 4382 |
| | 10 | 0.7 | $6.9 \pm 1.2$ | 4178 |
| | 10 | 1.0 | $10.4 \pm 2.1$ | 4182 |
| AFFINITYENHANCER (No Matching) | 1 | 0.7 | $1.2 \pm 0.4$ | 0 |
| | 1 | 1.0 | $2.3 \pm 1.1$ | 123 |
| | 5 | 0.7 | $1.1 \pm 0.4$ | 0 |
| | 5 | 1.0 | $2.4 \pm 1.1$ | 154 |
| | 7 | 0.7 | $1.2 \pm 0.5$ | 1 |
| | 7 | 1.0 | $2.4 \pm 1.2$ | 210 |
| | 10 | 0.7 | $1.2 \pm 0.5$ | 0 |
| | 10 | 1.0 | $2.6 \pm 1.2$ | 253 |
| AFFINITYENHANCER (No Embed) | 1 | 0.7 | $6.7 \pm 1.1$ | 703 |
| | 1 | 1.0 | $7.1 \pm 1.2$ | 2366 |
| | 2 | 0.7 | $26.2 \pm 2.1$ | 0 |
| | 2 | 1.0 | $28.4 \pm 2.6$ | 0 |
| AFFINITYENHANCER (CNN) | 1 | 0.5 | $10.1 \pm 1.5$ | 3645 |
| | 1 | 0.7 | $11.2 \pm 1.8$ | 3812 |
| | 1 | 1.0 | $14.7 \pm 2.3$ | 871 |
| | 2 | 0.5 | $32.6 \pm 1.7$ | 0 |
| | 2 | 0.7 | $32.9 \pm 2.0$ | 0 |
| | 2 | 1.0 | $34.8 \pm 2.5$ | 0 |
| | 3 | 0.5 | $42.8 \pm 1.6$ | 0 |
| | 3 | 0.7 | $43.6 \pm 1.9$ | 0 |
| | 3 | 1.0 | $45.9 \pm 2.4$ | 0 |
| AFFINITYENHANCER (No Adj) | 1 | 0.7 | $1.1 \pm 0.4$ | 0 |
| | 1 | 1.0 | $2.2 \pm 1.1$ | 109 |
| | 5 | 0.7 | $2.0 \pm 0.7$ | 0 |
| | 5 | 1.0 | $3.4 \pm 1.3$ | 754 |
| | 7 | 0.7 | $3.1 \pm 0.9$ | 76 |
| | 7 | 1.0 | $5.2 \pm 1.8$ | 2960 |
| | 10 | 0.7 | $6.7 \pm 1.4$ | 3848 |
| | 10 | 1.0 | $10.7 \pm 2.3$ | 3939 |

Table 10: Selecting sampling parameters for Seed 2 with the maximum number of designs within an edit distance of [5,12] from the seed sequence. We sampled 5000 designs for 3 internal seeds and Trastuzumab. AFFINITYENHANCER is the base model with a GearNet embedder and pOAS sequence decoder, an adjacency-informed Graph Transformer, and data matching. PropEn is the sequence-only model from Tagasovska et al. (2024).

| Model | Iterations | Temperature | Edit distance | # in [5,12] |
|---|---|---|---|---|
| AFFINITYENHANCER | 1 | 0.7 | $2.7 \pm 0.7$ | 3 |
| | 1 | 1.0 | $3.9 \pm 1.3$ | 1194 |
| | 5 | 0.7 | $5.4 \pm 1.0$ | 1293 |
| | 5 | 1.0 | $7.4 \pm 1.8$ | 4672 |
| | 7 | 0.7 | $7.5 \pm 1.2$ | 3515 |
| | 7 | 1.0 | $10.6 \pm 2.1$ | 4130 |
| | 10 | 0.7 | $13.3 \pm 1.7$ | 1619 |
| | 10 | 1.0 | $17.6 \pm 2.4$ | 48 |
| AFFINITYENHANCER (No Matching) | 1 | 0.7 | $1.4 \pm 0.6$ | 0 |
| | 1 | 1.0 | $2.8 \pm 1.3$ | 443 |
| | 5 | 0.7 | $1.5 \pm 0.6$ | 0 |
| | 5 | 1.0 | $3.0 \pm 1.4$ | 557 |
| | 7 | 0.7 | $1.6 \pm 0.6$ | 0 |
| | 7 | 1.0 | $3.1 \pm 1.4$ | 672 |
| | 10 | 0.7 | $1.8 \pm 0.7$ | 1 |
| | 10 | 1.0 | $3.4 \pm 1.5$ | 954 |
| AFFINITYENHANCER (No Embed) | 1 | 0.7 | $7.3 \pm 1.2$ | 1470 |
| | 1 | 1.0 | $7.6 \pm 1.3$ | 3486 |
| | 2 | 0.7 | $30.1 \pm 2.1$ | 0 |
| | 2 | 1.0 | $32.1 \pm 2.6$ | 0 |
| AFFINITYENHANCER (CNN) | 1 | 0.5 | $16.3 \pm 1.5$ | 10 |
| | 1 | 0.7 | $17.0 \pm 1.7$ | 13 |
| | 1 | 1.0 | $19.5 \pm 2.2$ | 2 |
| | 2 | 0.5 | $32.3 \pm 1.3$ | 0 |
| | 2 | 0.7 | $33.4 \pm 1.7$ | 0 |
| | 2 | 1.0 | $35.6 \pm 2.1$ | 0 |
| | 3 | 0.5 | $41.6 \pm 1.6$ | 0 |
| | 3 | 0.7 | $42.0 \pm 1.8$ | 0 |
| | 3 | 1.0 | $43.1 \pm 1.9$ | 0 |
| AFFINITYENHANCER (No Adj) | 1 | 0.7 | $1.8 \pm 0.5$ | 0 |
| | 1 | 1.0 | $3.0 \pm 1.2$ | 440 |
| | 5 | 0.7 | $5.0 \pm 0.9$ | 840 |
| | 5 | 1.0 | $6.6 \pm 1.6$ | 4353 |
| | 7 | 0.7 | $7.0 \pm 1.1$ | 2089 |
| | 7 | 1.0 | $9.1 \pm 1.8$ | 4769 |
| | 10 | 0.7 | $10.1 \pm 1.3$ | 3918 |
| | 10 | 1.0 | $13.6 \pm 2.2$ | 1547 |

Table 11: Selecting sampling parameters for Seed 3 with the maximum number of designs within an edit distance of [5,12] from the seed sequence. We sampled 5000 designs for 3 internal seeds and Trastuzumab. AFFINITYENHANCER is the base model with a GearNet embedder and pOAS sequence decoder, an adjacency-informed Graph Transformer, and data matching. PropEn is the sequence-only model from Tagasovska et al. (2024).

| Model | Iterations | Temperature | Edit distance | # in [5,12] |
|---|---|---|---|---|
| AFFINITYENHANCER | 1 | 0.7 | $1.1 \pm 0.3$ | 0 |
| | 1 | 1.0 | $1.9 \pm 0.9$ | 19 |
| | 5 | 0.7 | $3.3 \pm 0.9$ | 78 |
| | 5 | 1.0 | $4.6 \pm 1.4$ | 2106 |
| | 7 | 0.7 | $4.7 \pm 1.0$ | 1021 |
| | 7 | 1.0 | $6.5 \pm 1.7$ | 4352 |
| | 10 | 0.7 | $7.2 \pm 1.4$ | 4066 |
| | 10 | 1.0 | $10.6 \pm 2.2$ | 4062 |
| AFFINITYENHANCER (No Matching) | 1 | 0.7 | $1.1 \pm 0.3$ | 0 |
| | 1 | 1.0 | $1.9 \pm 0.9$ | 39 |
| | 5 | 0.7 | $1.2 \pm 0.4$ | 0 |
| | 5 | 1.0 | $2.1 \pm 0.9$ | 37 |
| | 7 | 0.7 | $1.2 \pm 0.4$ | 0 |
| | 7 | 1.0 | $2.1 \pm 1.0$ | 54 |
| | 10 | 0.7 | $1.3 \pm 0.5$ | 0 |
| | 10 | 1.0 | $2.3 \pm 1.0$ | 98 |
| AFFINITYENHANCER (No Embed) | 1 | 0.7 | $6.9 \pm 1.1$ | 434 |
| | 1 | 1.0 | $7.3 \pm 1.3$ | 1992 |
| | 2 | 0.7 | $23.7 \pm 2.1$ | 0 |
| AFFINITYENHANCER (CNN) | 1 | 0.5 | $7.1 \pm 1.0$ | 834 |
| | 1 | 0.7 | $7.6 \pm 1.2$ | 2933 |
| | 1 | 1.0 | $9.4 \pm 1.8$ | 4719 |
| | 2 | 0.5 | $16.7 \pm 1.4$ | 3 |
| | 2 | 0.7 | $17.4 \pm 1.7$ | 6 |
| | 2 | 1.0 | $19.7 \pm 2.1$ | 1 |
| | 3 | 0.5 | $26.3 \pm 1.5$ | 0 |
| | 3 | 0.7 | $26.9 \pm 1.8$ | 0 |
| | 3 | 1.0 | $29.0 \pm 2.3$ | 0 |
| AFFINITYENHANCER (No Adj) | 1 | 0.7 | $1.1 \pm 0.4$ | 0 |
| | 1 | 1.0 | $2.0 \pm 0.9$ | 31 |
| | 5 | 0.7 | $1.9 \pm 0.8$ | 0 |
| | 5 | 1.0 | $3.3 \pm 1.4$ | 747 |
| | 7 | 0.7 | $3.2 \pm 1.1$ | 199 |
| | 7 | 1.0 | $5.4 \pm 1.9$ | 3274 |
| | 10 | 0.7 | $7.2 \pm 1.7$ | 4423 |
| | 10 | 1.0 | $11.3 \pm 2.5$ | 3466 |

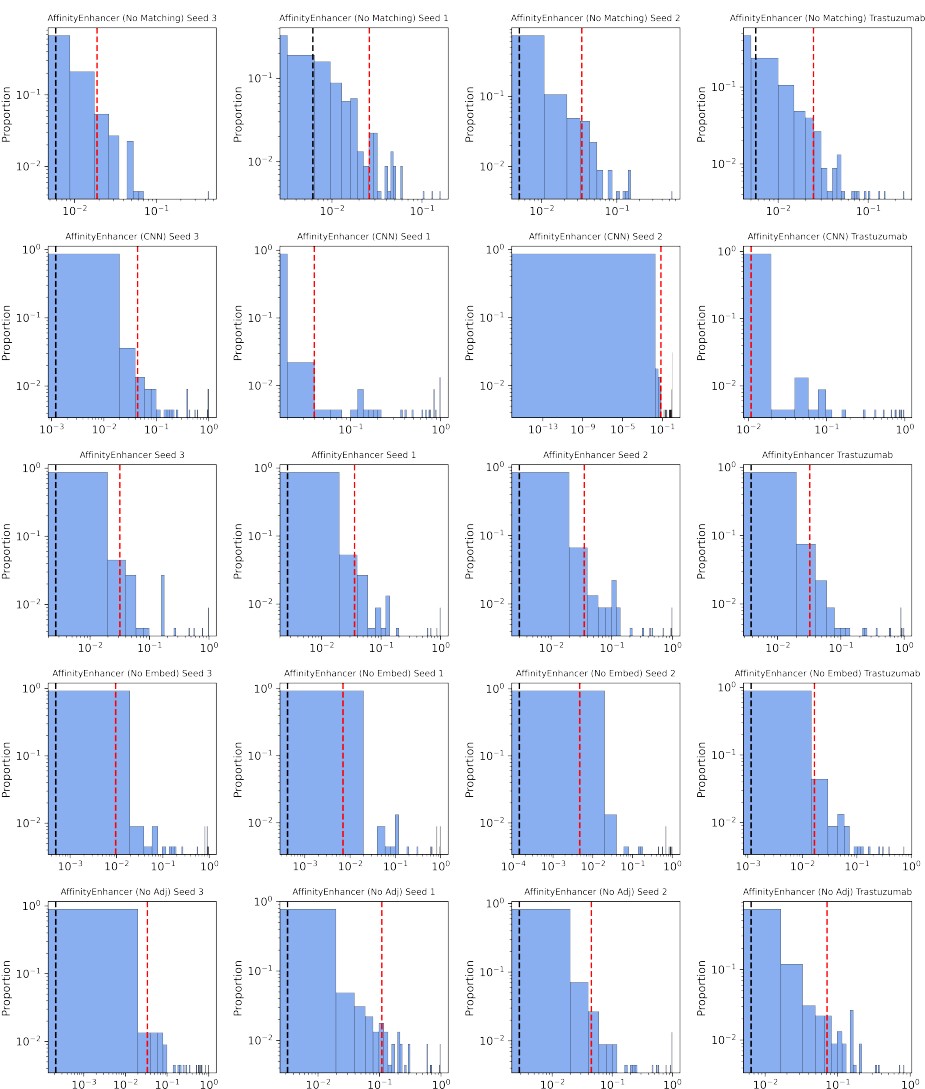

Figure 9: Distribution of fraction of designs with edits per-residue for each model and seed. Black and red dashed lines mark the 50th and 90th percentile respectively.

