# OpenReview forum: "Distilling Causal Signals for One-Shot Directed Evolution of Antibodies"
_ICLR.cc/2026/Conference — ICLR 2026 Poster_

### Official Review · Reviewer_Mw4x · 2025-10-27

**Soundness:** 3
**Presentation:** 3
**Contribution:** 3
**Rating:** 6
**Confidence:** 3

**Summary:**

This paper presents AFFINITYENHANCER, a one-shot framework for antibody affinity improvement: given a single lead antibody sequence, the method learns residual mappings in a structure-aware embedding space from matched (low-affinity, high-affinity) pairs collected under the same antigen environment, and decodes improved candidates. The authors report considerable in-silico gains against other baseline methods on several antibody seeds, including Trastuzumab.

**Strengths:**

- The one-shot affinity-maturation problem addressed in this paper is practical and challenging in therapeutic antibody development. In real-world discovery scenarios, we frequently have only a single “lead” sequence and lack complex structural data or large matched datasets.
- The authors propose a novel, matching-based computational framework for antibody affinity optimization that learns from matched low to high affinity pairs. This training strategy avoids relying on antigen–antibody complex structures during training and appears capable of improving performance in low-data antigens.
- In the reported in-silico evaluations, the proposed model outperforms baseline methods on several held-out antibodies.

**Weaknesses:**

Major
- Most of the reported improvements are evaluated using the Cortex predictor alone. Cortex itself is a deep-learning model and may introduce biases in the evaluation. The authors could supplement their validation with some other state-of-the-art predictors to provide a more comprehensive and robust assessment.
- The method depends on predicted structures (ESMFold). Antibody structure prediction, especially for the highly variable CDR loops, remains a difficult task. If the predicted structure is inaccurate, the residual mapping may learn an incorrect transformation. The paper currently lacks a failure analysis or robustness test for this critical scenario.

Minor
- Line 48: miniscule -> minuscule

**Questions:**

- The caption for Table 1  states: "wet-lab positives...". Please clarify: How was this "wet-lab positive" data obtained? Were any wet-lab experiments actually performed as part of this study?
- Did the authors try using sequence-only inputs (i.e., no predicted structure) for the model? Since predicted antibody structures are often inaccurate.

---

> ### Author Response · Authors · 2025-11-25
>
> We thank Reviewr Mw4x for highlighting the practical relevance of the one‑shot setting and for the constructive suggestions.
>
> **[Supplement validation with other state-of-the-art predictors to provide a more comprehensive and robust assessment.]**
> Please kindly read A in our common response where we compar Cortex to other publicly available state-of-the-art (SOTA) methods and found limited room for improvement: current methods perform on par with or worse than Cortex. Moreover, binding affinity prediction remains an open research problem, as accurate in silico prediction is still highly challenging [1, 2, 3].
>
> **[What happens if the predicted structure is inaccurate, lacking robustness test for this critical scenario]**
> This is an excellent point. In the previous version of the manuscript we omitted an important detail: for regularization and robustness, we add small Gaussian noise to the predicted structures during training. This practice was originally proposed in ProteinMPNN [4], where training with backbone noise was shown to improve protein design performance. In our experiments, adding noise similarly improves robustness and reduces overfitting to a specific folding method.
>
> Our ablations and visualizations also provide indirect evidence of robustness:
>
> Ablations with weakened structural priors.
> - Table 1 reports the “AffinityEnhancer without Adjacency (–Adjacency)” and “AffinityEnhancer without Embedding (–Embedding)” variants. Removing adjacency (fully connected graph transformer) degrades controllability and binder yields, but the model still produces improved binders for most seeds. This suggests that the structural prior is beneficial but not brittle for some of the seeds.
>
> - Predicted vs. experimental structure.
> For Trastuzumab, Figures 2 and 5 visualize edits on an ESMFold-predicted structure aligned to the crystal structure of the Trastuzumab–HER2 complex. Edits concentrate on rim residues contacting the antigen while avoiding the core, consistent with preserving the known binding pose even when starting from a predicted structure.
>
> We will make these robustness observations explicit in Sec. 5 and add a brief discussion clarifying that, in practice, even imperfect structures can provide useful relational information without requiring atomic-level accuracy.
>
> **[Caption for Table 1]**
> We apologize for the confusion. We do not have wet-lab validation yet. In Table 1, “wet-lab positives” was intended to indicate that Cortex was trained on wet-lab data. We have corrected the caption accordingly in the updated manuscript.
>
> **[Sequence-only inputs for the model]**
> Thank you for this suggestion. We trained a sequence-only variant of the model, using the same training set as the structure-based AffinityEnhancer. The results (included in Table 1 in the pdf) confirm the importance of structural information for binding affinity: the sequence-only model yields a 0% binding rate and 0% improved designs.
>
> Qualitatively, most mutations  of the sequence-only concentrate in the framework region, which carries little information about the binding interaction between the antibody and its target protein, explaining the lack of functional improvements.
>
> | Model                |   ED | ED window | Binders | Improved | Binder rate | Improved rate | Seeds improved (T + 1–3) |
> | -------------------- | ---: | --------: | ------: | -------: | ----------: | ------------: | -----------------------: |
> | AffinityEnhancer     | 7.08 |     4,555 |   2,505 |      423 |      50.10% |         8.46% |                      4/4 |
> | Sequence-only | 55.8 |         0 |       0 |        0 |        0.0% |         0.00% |                      0/4 |
>
>
> References:
>
> [1]Hummer, A.M., Schneider, C., Chinery, L. and Deane, C.M., 2025. Investigating the volume and diversity of data needed for generalizable antibody–antigen ΔΔ G prediction. Nature Computational Science, 5(8), pp.635-647.
>
> [2] Barlow, K.A., Ó Conchúir, S., Thompson, S., Suresh, P., Lucas, J.E., Heinonen, M. and Kortemme, T., 2018. Flex ddG: Rosetta ensemble-based estimation of changes in protein–protein binding affinity upon mutation. The Journal of Physical Chemistry B, 122(21)
>
> [3] Pires, D.E. and Ascher, D.B., 2016. mCSM-AB: a web server for predicting antibody–antigen affinity changes upon mutation with graph-based signatures. Nucleic acids research, 44(W1), pp.W469-W473.
>
> [4] Dauparas, J., Anishchenko, I., Bennett, N., Bai, H., Ragotte, R.J., Milles, L.F., Wicky, B.I., Courbet, A., de Haas, R.J., Bethel, N. and Leung, P.J., 2022. Robust deep learning–based protein sequence design using ProteinMPNN. Science, 378(6615).

---

### Official Review · Reviewer_WQqy · 2025-11-07

**Soundness:** 2
**Presentation:** 2
**Contribution:** 2
**Rating:** 2
**Confidence:** 5

**Summary:**

The paper proposes AFFINITYENHANCER, a one-shot framework for antibody affinity maturation without requiring antigen structure or fine-tuning on target-specific data. The model learns to enhance antibody binding by training on matched pairs of antibody sequences with known differences in binding affinity, leveraging structure-aware embeddings (GearNet) and a Graph Transformer that maps low-affinity to high-affinity embeddings. The approach is motivated by the lack of antigen-specific data and aims to generalize to unseen antibody seeds through causal learning signals derived from matched improvements across multiple environments. Experimental results on public datasets reportedly show that the method outperforms existing structure-conditioned and sequence-based baselines in improving affinity in silico.

**Strengths:**

- Proposes a clear formulation of "one-shot affinity maturation", which addresses an under-explored scenario.

- The matching-based learning framework is well explained and connected to causal inference and preference learning.

- Integration of sequence–structure embeddings and a residual graph transformer is technically sound and leverages state-of-the-art pretrained models.

- The paper includes some theoretical grounding for causal signal isolation in matched-pair training.

**Weaknesses:**

- Limited novelty compared to previous works such as "Protein Design by Directed Evolution Guided by Large Language Models" (IEEE TEVC 2025) [1] and "LatentDE: latent-based directed evolution for protein sequence design" (MLST 2025) [2]. Both prior studies also model directed evolution or affinity optimization using pretrained representations and search in learned latent spaces; the main difference here is merely the inclusion of structure-aware embeddings.

- The "one-shot" setting is more of a data-split protocol than a conceptual advance: similar generalization settings were already evaluated in LatentDE and related MLDE pipelines.

- No experimental validation (either wet-lab or cross-docking) to confirm the claimed affinity improvement; results are purely in silico.

- The methodological overlap with Property Enhancer (PropEn, 2024) is substantial, with only minor modifications (structure-conditioned matching).

- The causal theory section adds mathematical formality but limited biological insight or empirical validation.

*** References:

[1] Trong Thanh Tran and Truong-Son Hy, Protein Design by Directed Evolution Guided by Large Language Models, IEEE Transactions on Evolutionary Computation, vol. 29, no. 2, pp. 418-428, April 2025, DOI 10.1109/TEVC.2024.3439690.
URL: https://ieeexplore.ieee.org/document/10628050

[2] Thanh V. T. Tran, Nhat Khang Ngo, Viet Thanh Duy Nguyen, and Truong-Son Hy, LatentDE: Latent-based Directed Evolution for Protein Sequence Design, Machine Learning: Science and Technology, Volume 6, Number 1, DOI 10.1088/2632-2153/adc2e2.
URL: https://iopscience.iop.org/article/10.1088/2632-2153/adc2e2/pdf

**Questions:**

How does your model differ in practice from prior ML-based directed evolution frameworks (e.g., LatentDE or LLM-guided directed evolution) beyond adding structure embeddings? Specifically, what unique biological or algorithmic insights does AFFINITYENHANCER provide that cannot be replicated by those prior approaches?

---

> ### Author Response · Authors · 2025-11-25
>
> We appreciate Reviewer WQqy’s careful reading and respond to the main concerns below.
>
> **[Limited novelty compared to previous works such as MLDE [1] and LatentDE[2]]** Thank you for this remark. We agree that MLDE and LatentDE are important precedents in sequence-based directed evolution. However, our setting and approach differ in several key aspects.
>
> First, the problem setting is different: [1, 2] optimize sequence-only properties of a protein without explicit conditioning on a binding partner. In contrast, our focus is antibody–antigen affinity enhancement, where antibodies bind conditionally on the target’s surface. This conditionality makes structural information about the antibody central to the problem.
>
> In AffinityEnhancer (AE) we therefore combine:
>
> 	- the implicit matched-pair framework (inspired by PropEn),
>
> 	- an explicit structure-aware embedder
>
> 	-a structure-aware reconstructor that produces localized edits.
>
> Our ablations show that this combination is crucial: removing structural embeddings or matching causes a substantial drop in binding and improvement rates (Table 1), bringing performance closer to sequence-only baselines such as PropEn and MLDE-style approaches.
>
> To address the reviewer’s suggestion more directly, we additionally ran MLDE on Seed 1 and Trastuzumab in our setting. **Results have been added to the manuscript in Section H, Table 14.** The preliminary results in Table 14, section H, in the OOD case, an oracle-guided approach does not translate effectively to affinity enhancement, which is inherently target-dependent.
>
> Due to time constraints during the discussion period, we ran MLDE for two seeds, however, we plan to apply MLDE for all seeds and add the rest of the analysis in the camera ready version.
> We now also include the directed-evolution methods in the updated manuscript Table 6 (Positioning of AE relative to SOTA methods) for clarity and self-containment, as well as brief discussion in Section 2, Related Work.
>
> **[unique biological or algorithmic insights of AE]**
> Conceptually and practically, AE differs from MLDE and LatentDE in three main ways:
>
> 1. Problem setting: one-shot, target-agnostic training.
> LatentDE and MLDE pipelines are designed for target-specific optimization: they assume access to many labeled variants for a given target and train a property model or latent search procedure per target.
>
> 2. Supervision via environment-controlled, locally matched improvements.
> PropEn and related latent-search methods operate on global property gradients or black-box optimization, without explicit antigen conditioning.
> Our matching construction pairs sequences within the same antigen environment under tight distance and improvement thresholds (d_x, d_y), and learns a residual map in embedding space that is theoretically dominated by causal affinity-changing directions (Theorem 1 and Appendix B).
> This environment-controlled matching is key to distilling antigen-invariant, causal affinity-maturation modes.
>
> 3. Antibody-specific, structure-aware residual operator with targeted edits.
> As shown in Figure 2 and Table 1, AE proposes compact, CDR-localized edits concentrated on interface “rim” residues rather than the core. LatentDE-type frameworks do not explicitly enforce such structured, interface-aware edits.
>
> Empirically, these differences matter: in Table 1, the PropEn sequence-only baseline fails to generate binders near the seed (all designs are >25 edits from the parent).
>
> To summarize:
>
> __Algorithmic__: environment-controlled matching + a structure-aware residual operator yield a practical train-once, apply-many framework for antibody affinity maturation in the one-shot regime.
>
> __Biological__: the model discovers a consistent strategy of editing rim/interface residues while preserving the core binding interface, across distinct antibodies and without antigen structures or explicit epitope labels, suggesting that AE has internalized general principles of affinity maturation.
>
> **[one-shot setting is more of a data split protocol]** We believe there may be a misunderstanding. In our context, one-shot refers not only to a held-out family of proteins but also to the absence of target-specific labels and structures for the test antigen.
>
> **[No experimental validation]**
> Please kindly read A in our common response to all reviewers.
>
> **[Methodological overlap with Property Enhancer]**
> The only substantive overlap with PropEn is the use of implicit guidance via training on matched pairs.
>
> **[Limited biological insight/experimental validation of the causal analysis]We believe the reviewer may have missed Fig. 2 and Fig. 5, which highlight AE’s proposed mutations and show that it consistently identifies structurally distinct and relevant positions for each antibody.
>
> Theorem 1 connects directly to these biological observations in Sec. 5: for Trastuzumab and internal seeds, matched models concentrate edits on protruding CDR motifs and interface rims (Fig. 2 and Fig. 5).

---

### Official Review · Reviewer_trYu · 2025-11-12

**Soundness:** 3
**Presentation:** 3
**Contribution:** 3
**Rating:** 6
**Confidence:** 4

**Summary:**

The paper introduces AFFINITYENHANCER, a machine learning framework for one-shot affinity maturation of antibodies without requiring antigen or complex structural data. It builds on the concept of matched data (pairs of antibodies with measured affinity differences) and learns a residual mapping in embedding space from lower- to higher-affinity antibodies. The model uses GearNet for structure–sequence embeddings, a graph transformer to model causal affinity-enhancing directions, and a decoder trained on antibody sequence data. Experiments demonstrate strong in-silico improvements compared to baselines (PropEn, AntiFold, IgCraft) on SKEMPI-derived datasets and Trastuzumab.

**Strengths:**

Clearly formalizes one-shot affinity maturation, which is both scientifically and computationally relevant.
Provides a theoretical causal justification for why “matched improvements” capture meaningful binding signals.

Introduces a structure-aware residual map leveraging graph transformers and pretrained geometric embeddings.
The causal analysis (Theorem 1) connects local smoothness and bounded spurious movement to matched-pair supervision.

Comprehensive comparisons and ablations (matching, embedding, adjacency, transformer vs CNN).
Evaluations on multiple seeds, including a public antibody (Trastuzumab).
Consistent improvements over strong baselines.

The motivation and positioning against sequence-only and structure-conditioned baselines are clearly articulated.
The one-shot constraint is well justified with biological and data-availability arguments.

**Weaknesses:**

Experimental validation limited to in-silico predictions.
The study depends entirely on oracle models (Cortex) for affinity estimation.
Some discussion of potential wet-lab verification or validation strategy would strengthen impact.

Generalization claim could be quantified better.
“One-shot” evaluation is convincing qualitatively, but out-of-distribution metrics or diversity analyses are limited.
No discussion of potential failure cases (e.g., antibodies far from training distribution).

Complexity and reproducibility.
The model stack (GearNet + decoder + graph transformer) is nontrivial; ablation on computational cost or scalability would help.
The reproducibility section promises artifact release, but details like parameter counts per component or training time are missing.

**Questions:**

1.	Have you evaluated how the model performs on antibodies with significantly different scaffolds or germlines (i.e., distribution shift)?
2.	How sensitive is the method to the quality or number of matched pairs — could noisy or synthetic matches degrade performance?
3.	Could the model be extended to incorporate weak antigen information, e.g., paratope–epitope contact priors?
4.	Do you have plans for experimental or in-vitro validation to verify that the predicted affinity gains translate to real-world improvements?

---

> ### Author Response · Authors · 2025-11-25
>
> We appreciate Reviewer R1’s positive evaluation of the soundness, causal formulation, and empirical comparisons.
>
> **[W1. and Q4. - Experimental validation limited to in-silico predictions.]**
> We agree with this general remark and address it in the common response (Point A). Briefly:
> (i) we show that Cortex’s performance is on par with or surpasses publicly available SOTA predictors;
> (ii) wet-lab validation is currently in progress (∼2.5 months per round), and we expect to have results in time for the camera-ready version;
> (iii) in the meantime, we will include an analysis __in the next couple of days__ comparing mutations proposed by AffinityEnhancer (AE), to those of a dataset of experimentally characterized single-point mutation designs. This analysis is supported by prior work on epistasis, where there is broad agreement that, when the total number of mutations is small (typically <5–6), the effect of multiple substitutions can often be approximated as an additive or weakly epistatic combination of individual mutation effects [1]. Under this regime, the comparison provides a meaningful proxy for the quality of AE’s multi-mutation designs, and we observe that AE, relative to baselines, suggests structurally meaningful and empirically effective mutations.
>
> [1] Chen, A., Stanton, S.D., Ding, F., Alberstein, R.G., Watkins, A.M., Bonneau, R., Gligorijević, V., Cho, K. and Frey, N.C., 2024. Generalists vs. Specialists: Evaluating LLMs on Highly-Constrained Biophysical Sequence Optimization Tasks. ICML 2024.
>
> **[W2. and Q1. - Generalization/out-of-distribution metrics and distribution shift]**
> We agree that explicit OOD characterization is important, and we expand on this in common response B and in the updated Section F. The new tables summarizes quantitative OOD metrics, including distance to germlines and different scaffolds.
>
> **[W3. Complexity and reproducibility.]**
> We apologize for not including full model and resource details in the initial submission. We now add the following information in Section C and Tables 2, 3 and 4 in the Supplementary of the updated manuscript.
>
> **[Q2. How sensitive is the method to the quality or number of matched pairs — could noisy or synthetic matches degrade performance?]**
>
> Our theoretical analysis (Sec. 3 and Appendix B) shows that, under our matching rule (small d(x, x'), positive improvement Δy in the same environment), each pair induces a minimum causal movement and a bounded spurious movement in embedding space, biasing learning toward causal affinity-changing directions.
>
> Empirically, Table 1 compares AffinityEnhancer to the “–Matching” ablation, which effectively replaces guided improvements with autoencoding. The –Matching variant produces substantially fewer binders and improved binders, highlighting the importance of informative matches.
>
> These results indicate that AE is indeed sensitive to match quality: the signal from the pairing implicitly guides the gradient for each seed. To mitigate the impact of noisy labels, we recommend using an improvement threshold larger than the measurement noise. For binding affinity, after consulting with domain experts we chose a threshold of 0.5 (corresponding to a 3.16× affinity change).
>
> In principle, synthetic matches could be incorporated as well, again with careful curation to avoid uncertain predictor regions. We have not yet explored this direction, but we thank the reviewer for the suggestion—it is particularly appealing for low-data regimes such as binding affinity.
>
> **[Q3. Could the model be extended to incorporate weak antigen information, e.g., paratope–epitope contact priors?]**
> Yes, our framework can naturally incorporate weak antigen information. For example, one could augment the reconstruction module with:
>
> - residue-wise features encoding predicted paratope probabilities or interface propensities on the antibody, and/or
> - coarse epitope embeddings for the antigen when available.
>
> These quantities would enter as additional node/edge features for the Graph Transformer. The matching scheme and causal analysis remain unchanged, as they already condition on fixed environments.
>
> In the current work, __we focus on the practically important setting where no antibody–antigen structural information is available at design time__. Extending AE to include paratope/epitope information is a natural and promising direction, and we will add a short discussion of this in Sec. 6.

---

### Author Response · Authors · 2025-11-25
**Response to all reviewers**

We thank all reviewers for their thoughtful and constructive feedback.

Below we first address common concerns:

**A. Experimental and in-silico validation (Reviewers trYU, WQqy, mW4x)**

__[Prospective wet-lab validation (in progress).]__ We fully agree that experimental validation is essential. We have already started a wet-lab campaign in which, for each held-out seed, we select a panel of top-ranked designs from AffinityEnhancer and appropriate baselines and test them in binding assays (with positive / negative controls and replicate measurements). These experiments will not complete during the rebuttal period but are on track for inclusion in the camera-ready version, where we will report hit rates and fold improvements relative to the seed for each method.

In parallel, we will add a retrospective analysis (Section XX) comparing mutations proposed by AffinityEnhancer (AE) to a dataset of experimentally characterized single-mutation (mutagenesis) designs.

__[In-silico validation - Cortex achieves state-of-the-art in-distribution prediction performance]__
All training labels in our matched dataset come from experimentally measured affinities in SKEMPI 2.0 and internal campaigns (pKD), as described in Sec. 5 and Appendix C–D.

For prospective designs, we deliberately use Cortex as a state-of-the-art in-silico oracle to avoid over-claiming experimental results for thousands of variants. **The seeds for prospective designs were chosen such that, for each “blind” or test seed, Cortex’s training set contains a sufficiently large number of labeled affinity examples.** This setup helps ensure that Cortex’s affinity predictions for our prospective designs are as accurate and well-calibrated as possible in an in-distribution regime.

Against these baselines, **Cortex delivers state-of-the-art in-distribution affinity prediction**: 82.9% mean binder accuracy (SE 0.4%) and Spearman \(\rho = 0.90\) for pKD regression, with per-target accuracies of 72.4% (Seed 1), 62.0% (Seed 2), and 78.4% (Trastuzumab). Further details are provided in Tables 2 and 3 in Section D of the updated manuscript.


**B. Generalization / “one‑shot” and out‑of‑distribution seeds (Reviewers trYU, WQqy)**

Our experimental setup is designed to enforce a **strict one-shot, out-of-distribution** regime:

1. **Held-out seeds are far from training data.**
   Tables 7 and 8 (Appendix G) quantifies the edit distance and germline overlap of each seed to the matched training set, separately for each CDR and framework region. All four seeds lie well outside the bulk of the training distribution (often >10–40 mutations away from the nearest training antibody in key CDRs).

2. **No target-specific fine-tuning or seed-adjacent supervision.**
   For each test seed, all sequences within a neighborhood of that seed are removed from the training matched dataset, and the model is never fine-tuned on the seed or its antigen. This contrasts with the usual protein optimization setups, where a model is explicitly fit on seed-adjacent variants for each target, and optimization is then performed in that locally supervised regime. Such approaches require a discriminative model for guidance, which in the case of binding affinity has a limited benefit. However, predicting antibody-antigen binding affinity is profoundly challenging for machine learning models, particularly in Out-of-Distribution (OOD) settings, making property oracles (like those used in directed evolution approaches) highly inaccurate.

**C. Rebuttal summary and manuscript updates**

In response to the reviewers’ comments, we have made the following additions and clarifications in the revised manuscript (marked in red in the pdf):

1. **In-silico predictor accuracy and regression metrics.**
   We report full accuracy and pKD regression metrics for our binding predictior - Cortex in new Tables 2 and 3 (Sec. D).

2. **Out-of-distribution metrics for seeds.**
   We added tables (Tables 7 and 8) quantifying OOD statistics for each seed relative to the matched training set.

3. **New directed-evolution, sequence-only baseline (as suggested by Reviewer WQqy).**
   We implement and evaluate a directed-evolution baseline from (Tran et al. 2024).

4. **Retrospective wet-lab–validated mutation analysis (proxy analysis while prospective assays are in progress).**
   We will include an analysis (Section XX) comparing AE’s proposed mutations to a dataset of experimentally validated single-mutation designs.

5. **Reproducibility and scalability details.**
   We provide detailed model and resource information, including parameter counts per component, training configurations (dataset sizes, GPUs, wall-clock times), and inference cost.

6. **Evaluation on an additional in-silico proxy task (ΔΔG).**
   We introduce an evaluation on predicted ΔΔG as a complementary in-silico proxy for binding affinity (details to be added; tba), further probing AE’s ability to capture affinity changes.

---

### Author Response · Authors · 2025-12-02
**Summary of changes**

Below we summarize how the revised manuscript (red markup in the pdf) addresses the main concerns raised by the reviewers.

| Reviewer Concern                                                                                       | Clarification / New Evidence                                                                                                                                                                                                                                                                                                                                                                                                                               | Action in Revised Manuscript                                                                                                                                                                          |
| ------------------------------------------------------------------------------------------------------ | ---------------------------------------------------------------------------------------------------------------------------------------------------------------------------------------------------------------------------------------------------------------------------------------------------------------------------------------------------------------------------------------------------------------------------------------------------------- | ----------------------------------------------------------------------------------------------------------------------------------------------------------------------------------------------------- |
| **C1. Reliability of the in‑silico oracle and unclear meaning of “binders”.**                          | Report full performance of Cortex                                                        | New Appendix **D. Affinity Oracle**, including **Tables 5–6**, and updated wording around **Table 1**.                                                                                                |
| **C2. Are “one‑shot” seeds truly out‑of‑distribution (OOD)? Possible train–test leakage.**             | Quantify OOD for every seed by reporting heavy, light, full‑length, and per‑CDR edit distances between each seed and the matched training set (all four design seeds are **64–87 aa** away; 60–70% sequence identity). Also report overlap of V/J germline genes.                                                                            | Added **Tables 7–8** with OOD statistics in Appendix F and new text at the start of **Sec. 5.1** highlighting that all evaluation seeds are far OOD from the training set.                            |
| **C3. Limited novelty vs MLDE / LatentDE and missing directed‑evolution baselines.**                   | Implemented an oracle‑guided latent directed‑evolution baseline (MLDE) for Seed 1 and Trastuzumab. Across all tested configurations, **AffinityEnhancer produces more binders and more improved binders at comparable edit distances** than MLDE.                                     | New **Appendix H** with **Table 14 (“Comparison of AffinityEnhancer with MLDE”)** and **Table 13** (“Positioning of AE with respect to SOTA methods”). |
| **C4. Lack of experimental support and biological interpretability of edits.**                         | Performed a **retrospective comparison to experimental single‑mutant affinity data** for three seeds. **AE’s high‑incidence edit positions are enriched for mutations with strong experimental affinity gains and often include the best experimental positions;** remaining misses are explicitly analyzed. | New subsection in **Sec. 5** with **Figures 6–8** and accompanying text discussing how AE’s edits align with experimental measurements and interface geometry.                                        |
| **C5. Robustness to noisy predicted structures and need for sequence‑only / architectural ablations.** | A **sequence‑only variant** included in Table 1 (“PropEn / –Structure”) and multiple ablations. These controls show that removing structural embeddings or matching collapses binder and improvement rates (often to **0%**), while full AE concentrates edits in CDRs and interface‑proximal residues even when structures come from noisy ESMFold predictions.                                   | Updated **Table 1** and surrounding **Sec. 5** text with the sequence‑only baseline and expanded ablation analysis, including an explicit discussion of robustness to imperfect structures.       |
| **C6. Missing details on compute, parameters, and reproducibility.**                                   | Provide explicit parameter counts and training/inference costs.                                                                                                          | New **Appendix C. Model and Training** with **Tables 2–4** and descriptive text covering dataset sizes, GPUs, wall‑clock times, and inference throughput.                                             |

---

### Meta-Review · Area_Chair_mmQ2 · 2026-01-06

**Summary:**

This paper proposes AffinityEnhancer, a one-shot framework for antibody affinity maturation that learns from locally matched low- vs high-affinity antibody pairs collected under the same antigen environment. The method learns a residual operator in a structure-aware embedding space (GearNet embeddings + graph transformer) and decodes improved sequences without using antigen structures, epitope labels, or target-specific fine-tuning at test time.

Reviewers agree the problem setting is practically important and the approach is technically sound, with strong in-silico gains over several sequence-only and structure-conditioned baselines across multiple seeds (including Trastuzumab). The main disagreement is whether the contribution is sufficiently novel relative to prior directed evolution / latent-space optimization frameworks (e.g., MLDE, LatentDE, PropEn variants), and how much confidence to place in oracle-based evaluation without completed wet-lab validation.

**Reviewer Concerns:**

Addressed or partially addressed by the rebuttal / revised manuscript:

- Oracle clarity (Mw4x, trYu): Added an affinity-oracle appendix with Cortex performance and clarified what “binders” means, improving transparency though still oracle-based.

- One-shot and OOD rigor (trYu, WQqy): Added per-seed OOD statistics (edit distances, germline overlap) and described leakage controls, strengthening the claim of true one-shot, OOD evaluation.

- Novelty vs directed-evolution baselines (WQqy): Implemented an oracle-guided MLDE baseline (Seed 1, Trastuzumab) and reports AE improves binder and improved-binder counts at similar edit distances, reducing the “missing baseline” concern.

- Biological interpretability (WQqy, trYu): Added retrospective comparison to experimental single-mutant datasets, showing AE edit hotspots align with known beneficial sites and analyzing misses.

- Reproducibility and compute (trYu): Added parameter counts, training/inference costs, and implementation details.

- Robustness to structure noise (Mw4x): Added sequence-only and architectural ablations showing large drops without structure/matching, plus discussion of robustness practices.

Outstanding / not convincingly addressed:

- Wet-lab validation: Prospective assays are still pending; core conclusions remain dependent on an in-silico oracle with possible bias under shift.

- Novelty: Concerns are mitigated but not fully resolved; some may still view the method as a well-executed integration rather than a distinct algorithmic advance.

**Reviewer Scores:**

trYu (6): Likely more confident after added oracle/OOD/compute clarifications.

Mw4x (6): Better addressed via ablations and robustness discussion; still limited by oracle-only evaluation.

WQqy (2->3 or stay the same): Key objections are engaged (MLDE baseline, clearer one-shot definition), but may not change the reviewer’s novelty threshold since the reviewer is very confident in the domain and mentioned that the proposed work overlaps quite a lot with existing work.

---

### Decision · Program_Chairs · 2026-01-26

Accept (Poster)